# Exploring Diverse In-Context Configurations for Image Captioning

**Xu Yang**[1]*, **Yongliang Wu**[1], **Mingzhuo Yang**[2], **Haokun Chen**[1], **Xin Geng**[1]

[1]School of Computer Science & Engineering, Key Lab of New Generation Artificial Intelligence
Technology & Its Interdisciplinary Applications (Ministry of Education), Southeast University
[2]The Chinese University of HongKong, Shenzhen
xuyang_palm@seu.edu.cn, yongliangwu@seu.edu.cn,
118020068@link.cuhk.edu.cn, chenhaokun@seu.edu.cn, xgeng@seu.edu.cn

## Abstract

After discovering that Language Models (LMs) can be good in-context few-shot learners, numerous strategies have been proposed to optimize in-context sequence configurations. Recently, researchers in Vision-Language (VL) domains also develop their few-shot learners, while they only use the simplest way, *i.e.*, randomly sampling, to configure in-context image-text pairs. In order to explore the effects of varying configurations on VL in-context learning, we devised four strategies for image selection and four for caption assignment to configure in-context image-text pairs for image captioning. Here Image Captioning is used as the case study since it can be seen as the visually-conditioned LM. Our comprehensive experiments yield two counter-intuitive but valuable insights, highlighting the distinct characteristics of VL in-context learning due to multi-modal synergy, as compared to the NLP case. Furthermore, in our exploration of optimal combination strategies, we observed an average performance enhancement of 20.9 in CIDEr scores compared to the baseline. The code is given in `https://github.com/yongliang-wu/ExploreCfg`.

## 1 Introduction

In contemporary times, the Language Model (LM) [1; 2] has emerged as a pivotal player in the field of Natural Language Processing (NLP). It accomplishes this by unifying a range of diverse NLP tasks into a shared **prompt paradigm** [3; 4]. To elaborate, a LM, unsupervised trained via conditional language modeling on a substantial volume of non-annotated data collected from the web, can reformulate various downstream tasks into fitting textual prompts. These prompts contain slots, the word probabilities of which are calculated using the pre-trained LM. This process obviates the necessity for gradient updates to the parameters of LM, thus mitigating the challenges associated with additional data collection and fine-tuning. To further enhance the effectiveness of this paradigm, the few-shot prompt (or in-context learning) [5] is proposed where a few examples are provided as additional contexts to guide the LM to generate the desired result.

Witnessing the success of the prompt paradigm in the NLP field, there has been a concerted effort by researchers to replicate its function in the Vision-Language Model (VLM). To facilitate this, Flamingo [6] is proposed to align well-trained large-scale vision and language models through some trainable cross-modal adapters. Consequently, the resultant model is capable of addressing Vision-Language (VL) tasks by processing a prompt sequence, which includes several interleaved image and text examples for in-context learning. Since the primary objective of Flamingo is to build a VLM for the few-shot prompt, they only apply a straightforward strategy to configure the in-context

---

*Corresponding author

37th Conference on Neural Information Processing Systems (NeurIPS 2023).

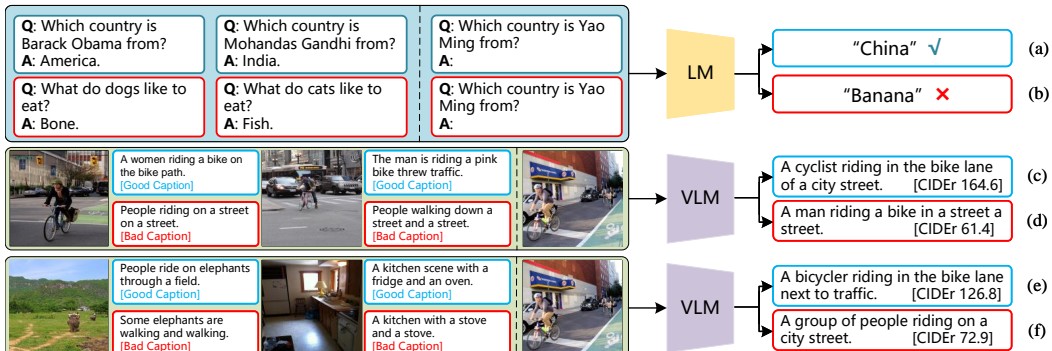

Figure 1: The distinction between LM and VLMs as few-shot learners. LM generally excel with examples akin to the test case (blue blocks in (a)). In contrast, for VLMs, the performance is not strictly correlated with image similarity but heavily relies on the caption quality. For instance, when low-quality captions are used, similar images (d) lead to worse performance than dissimilar ones (f) since VLMs may build a short-cut by reusing in-context captions without seeing the given images.

sequence by randomly sampling a few image-text pairs. Nevertheless, a plethora of studies within the NLP field [7] have demonstrated that diverse in-context configurations lead to dramatic effects on few-shot performance, *e.g.*, the selection or ordering of in-context samples [8; 9; 10], while only a limited number of studies systematically explore such effects in the VL case.

To narrow this gap, we explore the effects of various in-context configurations on the performance of few-shot VL tasks. Among various VL tasks, Image Captioning (IC) aims at generating a text conditioned on the source image, and thus can be considered as the visually-conditioned LM. Just as a multitude of NLP tasks can be recast as LM tasks, IC performs a similar function [11; 12; 13], which motivates our decision to select IC as the subject of our case study. However, unlike in NLP, where only single-modal texts are considered in the in-context configuration, in IC, the synergy between multiple modalities significantly influences the performance. For instance, our experiments revealed that selecting images similar to the test image for the in-context sequence does not always lead to good performance, a result closely tied to the quality of the associated captions. Figure 1 shows the comparison between LM and VLM as the few-shot learners.

Consequently, we design diverse ways to select images as the in-context images. After selection, the captions of different qualities are assigned to these images for constructing the multi-modal in-context sequence. By combining diverse image selection and caption assignment techniques, we undertake a comprehensive exploration of the effects of multi-modal mutual synergy on VL in-context captioning. We implement all experiments using the prevalent captioning dataset, MSCOCO [14], employing its training set as the database for image selection.

To select images, we use 4 different ways, which are Random Sampling, Similarity-based Image-Image Retrieval, Similarity-based Image-Caption Retrieval, and Diversity-based Image-Image Retrieval. Following image selection, various types of captions produced by 4 different strategies are assigned, which are Ground-Truth Captions, Model-Generated Captions, Iteratively Prompting, and Model-Generated Captions as Anchors.

Through extensive evaluation of various image selection and caption assignment strategies, we uncover two counter-intuitive yet valuable insights. (1) Caption quality is determined by descriptiveness and language patterns, but their influence on in-context captioning performance is unequal. When captions adequately describe salient image objects, simpler language patterns may yield better results. (2) The efficacy of similar images depends on the quality of the paired captions. Excessive similarity might cause VLM to create a short-cut inference [15] from in-context captions, potentially misleading the model with low-quality captions. Beyond these findings, we introduce a practical in-context captioning strategy, Iterative Prompting, for cases with limited or no Ground-Truth Captions. Furthermore, when Ground-Truth Captions are available, we recommend using Model-Generated Captions as anchors to identify which Ground-Truth Caption is a more suitable in-context caption. Experimental results indicate that even when utilizing low-quality model-generated captions, there is an average CIDEr improvement of 7.3. Moreover, in optimal conditions, the average enhancement reaches up to 20.9 points compared to the random sampling baseline.

Since the Flamingo code is not publicly available, our experiments mainly utilize the unofficial Open-Flamingo [16][2]. It's worth noting that the performance of Open-Flamingo is not on par with the official Flamingo due to its training on less data.

## 2   Related Work

**Prompting Language Model (LM).** The research paradigms of NLP have encountered two sea changes in the past few years [3]. The first one is the LM that is pre-trained by predicting the next word conditioned on observed words and can be fine-tuned for solving various downstream tasks, including GPT [17], BERT [1], and BART [18]. The second sea change is the emergence of the prompt paradigm, which was introduced with GPT-3 [2]. Within this paradigm, a pre-trained LM does not require fine-tuning to solve downstream tasks; instead, tasks are reformulated into appropriate prompts with empty slots to be filled. Subsequently, more advanced prompt-based techniques have been proposed, including prompt-tuning [19; 20; 21] and Chain-of-Thought [22; 23; 24; 25].

**Prompting Vision-Language Model (VLM).** In contrast to NLP, the Vision-Language (VL) domain has made significant strides in the first sea change, as evident by the development of various VL-BERT models. These models leverage large volumes of web-collected image-caption pairs to learn VL-generalizable embeddings [26; 27; 28; 29; 30; 31; 32; 33]. However, the prompt paradigm, despite revolutionizing NLP studies, only appears when a certain scale of the model is reached [34]. This scale prerequisite poses further challenges to the development of a VLM with prompt and in-context learning ability.

To mitigate training burdens, instead of updating all the parameters of a VLM [35; 36], some VLMs [35; 36; 37] freeze well-trained Language Models and only train a smaller network, referred to as adapters [38; 39; 40], to align the pre-trained vision and language models. Inspired by these models, both Frozen [41] and Flamingo [6] evolve into multi-modal few-shot learners by training vision and cross-modal adapters, respectively. Given its superior in-context learning ability, we use Flamingo to explore the effects of various in-context configurations [16].

There are also models that address VL tasks through in-context learning. For instance, PICa [11] utilizes captions as mediators to construct an in-context text for solving Visual Question Answering (VQA) tasks, while this may lose the mutual synergy in the representation space of different modalities. The models proposed in [42] and [36] both fine-tune VLMs for specific VQA tasks, but lack the generalized few-shot prompt ability for other VL tasks. UNIFIED-IO [43]demonstrates a unified approach to a myriad of tasks, from classical computer vision to natural language processing, without task-specific fine-tuning. And ProGrad [44] introduces a technique to prevent prompt tuning from forgetting pre-trained vision-language models' general knowledge by only updating aligned prompts, outperforming other methods in various few-shot learning scenarios.

**Exploring In-Context Configurations in NLP.** Upon observing that pre-trained LMs are good few-shot learners, researchers also discover that diverse in-context configurations have dramatic effects on performance [3]. This observation sparks numerous studies aimed at determining the optimal in-context configurations, such as the format of the in-context examples [45; 21; 5], the selection of these examples [8; 9; 46; 47], and even the order in which these examples are presented [48; 49; 10]. However, these studies are predominantly conducted within the NLP field and fail to consider the unique characteristics and complexity of multi-modal data. In order to address this gap, we propose a series of strategies to explore the effects of various multi-modal in-context sequence configurations.

**Image Caption.** Image Captioning (IC) [50] aims at correctly verbalizing one image using descriptive languages, which can be solved through both retrieve [51] and generation [50], the former retrieves complete sentences from a corpus, while the latter generates words sequentially. Researchers have recently combined these approaches by first retrieving image-caption pairs and inputting them into generation models [52; 53]. This process resembles in-context captioning, which also retrieves image-caption pairs to help captioning. However, in contrast to them [52; 53], our work introduces novel image and caption selection strategies to study in-context captioning and these methods can also enhance traditional retrieval-generation methods.

---

[2]Project: `https://laion.ai/blog/open-flamingo/`

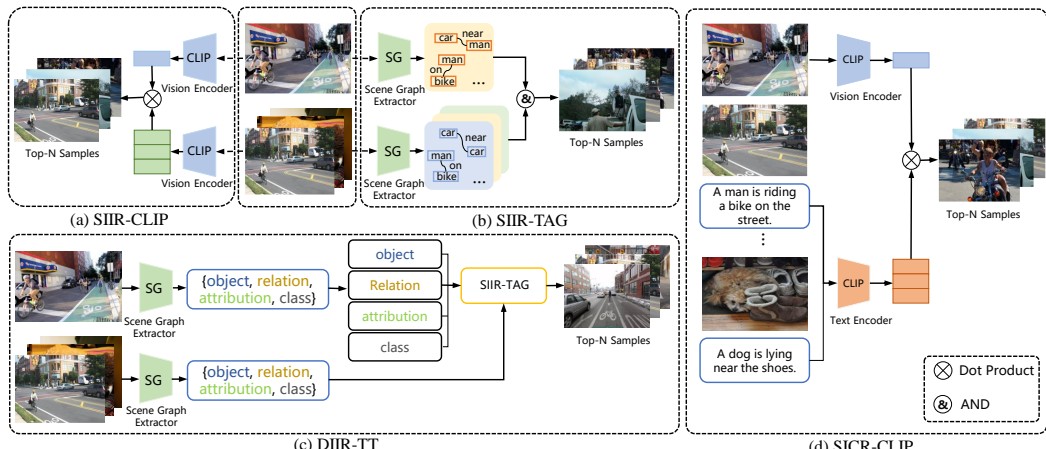

Figure 2: Image selection strategies: (a) SIIR-CLIP, (b) SIIR-TAG, (c) DIIR-TT, (d) SICR-CLIP.

# 3 Configuring In-Context Sequences

The in-context captioning can be treated as a vision-language conditioned text generation task. Given the multi-modal in-context sequence $\mathcal{S} = \{(\boldsymbol{I}_1, \boldsymbol{C}_1); (\boldsymbol{I}_2, \boldsymbol{C}_2); ...; (\boldsymbol{I}_n, \boldsymbol{C}_n); \hat{\boldsymbol{I}}\}$ that contains $n$-shot image-caption pairs $(\boldsymbol{I}, \boldsymbol{C})$ and one test image $\hat{\boldsymbol{I}}$, we hope to generate the corresponding image caption $\hat{\boldsymbol{C}} = \{\hat{w}_1, ..., \hat{w}_T\}$ in an auto-regressive manner. Here the $t$-th word $\hat{w}_t$ is sampled from the following word distribution:

$$P(\hat{w}_t | \mathcal{S}, \hat{w}_{1:t-1}), \tag{1}$$

where the probability $P(\cdot)$ is calculated by a pre-trained Vision-Language Model (VLM) (e.g., Flamingo [6] or Otter [54]).

Various studies in NLP [8; 45; 48; 46] field have shown that the performance of in-context learning varies significantly with different in-context configurations. We explore these effects in the case of Image Captioning (IC). Unlike pure NLP tasks, IC is a dual-modal task, which makes it more complex than NLP tasks. Specifically, the mutual synergy of image-caption examples must be considered in in-context learning, as opposed to considering the images or the captions independently. We next respectively introduce the image selection (cf. 3.1) and caption assignment (cf. 3.2) strategies used to configure in-context image-caption pairs.

## 3.1 Selecting Images

**Random Selection (RS).** Given a set $\mathcal{D} = \{(\boldsymbol{I}_1, \boldsymbol{C}_1), ..., (\boldsymbol{I}_M, \boldsymbol{C}_M)\}$ with $M$ image-caption pairs, we randomly sample $n$ images as $\{\boldsymbol{I}_1, ..., \boldsymbol{I}_N\}$ in $\mathcal{S}$.

**Similarity-based Image-Image Retrieval (SIIR).** Certain NLP studies suggest that performance can be enhanced by retrieving examples similar to the test case [8; 46; 47]. Following this approach, we retrieve $n$ images with the highest similarity scores to the test image $\hat{\boldsymbol{I}}$ from $\mathcal{D}$. We employ two methods to compute these similarities: 1) **SIIR-CLIP** (Figure 2 (a)): Using the vision encoder of CLIP [35], we extract image embeddings to determine similarities. 2) **SIIR-TAG** (Figure 2 (b)): We utilize some scene graph extractors to derive tags for each image. Specifically, we employ Vinvl [55] to extract objects and their attributes, and IETrans [56] to determine the relations present within the image. Following this extraction, we conduct an AND operation to compute the similarity between tags.

**Similarity-based Image-Caption Retrieval (SICR-CLIP)** (Figure 2 (d)). Taking advantage of the cross-modal retrieval capability of CLIP [35], we use its vision and language encoders to embed images and captions into a shared space for computing similarities. Given $\hat{\boldsymbol{I}}$, we calculate its cross-modal embedding similarities with $\{\boldsymbol{C}_1, ..., \boldsymbol{C}_M\} \in \mathcal{D}$, and select the images whose captions have top-$n$ similarities with $\hat{\boldsymbol{I}}$. Note that we use different kinds of captions as mediators; the methods to generate these captions will be detailed in Section 3.2. These captions are also used as $\boldsymbol{C}_1, ..., \boldsymbol{C}_N \in \mathcal{S}$.

**Diversity-based Image-Image Retrieval (DIIR)**. For a single test sample, beside using similar in-context examples, some NLP studies find that diversity is also crucial [57]. Consequently, we retrieve a diverse set of images from $\mathcal{D}$ to configure $\mathcal{S}$. Specifically, we employ the following strategies: 1) **DIIR-TR**: We extract discrete tags from the aforementioned VinVL and IETrans. Then we randomly divide the tags into $N$ clusters and apply SIIR-TAG to retrieve the most similar image from each cluster. 2) **DIIR-TT** (Figure 2 (c)): To conveniently control the number of shots, we incorporate class tags generated by the IETrans model, extending the basis of SIIR-Tags to four categories: object, class, attribute, and relation. We then employ SIIR-TAG to identify the top-$k$ similar images within each category, allowing us to create $4k$-shot images in $\mathcal{S}$. Note that both DIIR methods take into account a certain level of similarity during retrieval, rather than selecting entirely distinct images.

## 3.2 Assigning Captions

After selecting images, we need to assign one caption to each image to construct in-context image-caption pairs. To explore the effects of multi-modal mutual synergy, we use captions with diverse qualities as introduced in the following.

**Ground-Truth Captions (GTC).** In the MSCOCO dataset [14], each image has 5 GTCs and we simply use the first one in $\mathcal{S}$.

**Model-Generated Captions (MGC).** Compared with GTC, MGC has lower quality due to two disadvantages. Firstly, MGC uses poorer language, *e.g.*, simple words or rigid sentence patterns. Secondly, MGC has less descriptiveness that it may mis-verbalize or miss certain salient vision patterns of an image. However, we will see these two disadvantages do not equally cause worse performance compared with GTC in in-context captioning and surprisingly, we find that sometimes simple words or rigid sentence patterns even help generate good captions.

Here we apply two different models to generate the captions with diverse qualities. **MGC-TF@$X$:** a Transformer is trained from scratch by using VinVL features and $X$ denotes the CIDEr score [58] on the test set. To get captions of different qualities, we use the checkpoints from different training epochs and totally generate three kinds of MGCs. 1) MGC-TF@66 contains grammar mistakes while can describe the most salient objects. 2) MGC-TF@88 can use relatively correct grammar to describe the salient objects. 3) MGC-TF@135 is generated by a well-trained Transformer, *i.e.*, the loss converges. **MGC-VLM($N$)@$X$:** Another way to get MGCs is to use VLM in a $N$-shot manner. To achieve this, for each image $I \in \mathcal{D}$, we treat $I$ as the test image and then use Eq. (1) to generate a corresponding caption $C$ conditioned on $\mathcal{S}$ constructed by only $N$ image-caption pairs. As a result, two kinds of captions are got which are MGC-VLM(0)@63 and MGC-VLM(32)@81.

Both two ways can construct the set $\mathcal{D}$ with $M$ image-caption pairs. Then for a novel test image $\hat{I}$, we can select some image-caption pairs from $\mathcal{D}$ by some above-mentioned approaches, *e.g.*, using RS or SIIR-CLIP to get images and assigning the MGCs to the image, to configure $\mathcal{S}$ for generating a new caption. Compared with MGC-TF, MGC-VLM is more practical as it addresses scenarios where only a handful or even no human-labelled captions are available, which means we do not have enough data to train a Transformer from scratch.

**Iteratively Prompting (IP).** For MGC-VLM introduced before, one natural extension is IP. To achieve this, in the first iteration, we generate a caption for each image $I \in \mathcal{D}$ by MGC-VLM. In the subsequent iteration, these generated captions are paired with the selected images to prompt VLM to generate enhanced captions. This process can be repeated across multiple iterations, thereby iteratively prompting the VLM.

**Model-Generated Captions as Anchors (MGCA).** A MGC can serve not only as an in-context caption but also as an anchor for selecting a suitable caption from the five GTCs. As MGCs typically verbalize salient visual patterns in an image but may miss finer details, using them as anchors can lead to the selection of GTCs that highlight these salient patterns. Furthermore, the selected GTC may supplement interesting details about these patterns, potentially assisting VLM to generate superior captions during in-context learning. In the MGCA implementation, for each selected image, we measure the similarity between the MGC and five GTCs using CIDEr and select the GTC with the highest CIDEr score.

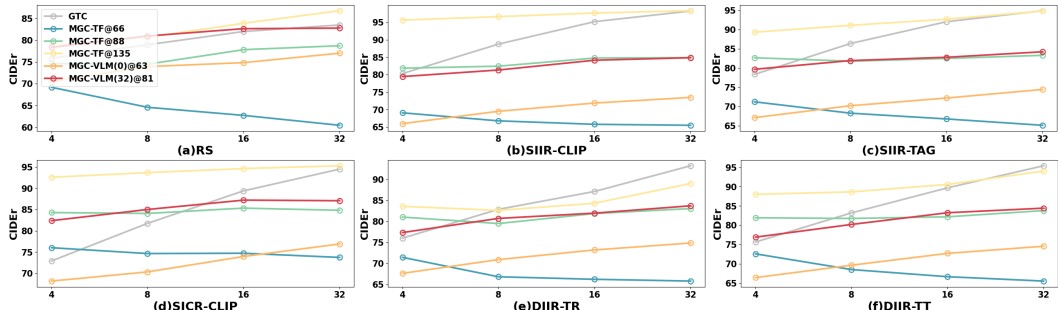

Figure 3: The line charts of various in-context captions with diverse image-selection strategies.

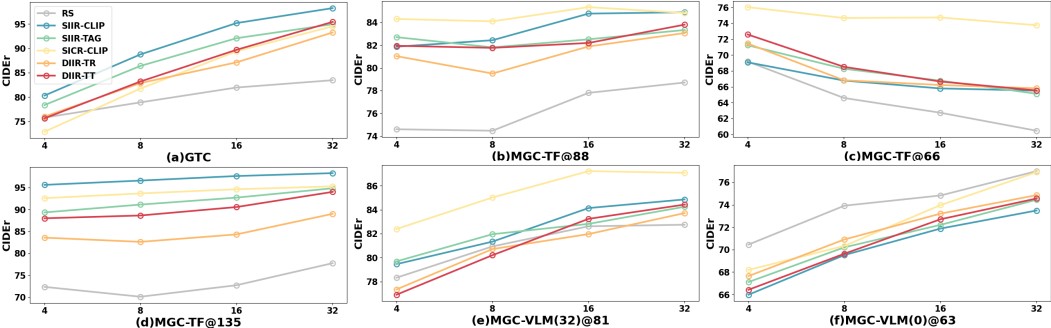

Figure 4: The line charts of various in-context images with diverse caption-assignment strategies.

## 4 Experiments

### 4.1 Dataset and Implementation Details

**MSCOCO.** We evaluate the proposed strategies on MSCOCO dataset [14], which is the most widely used benchmark in image captioning. We used the Karpathy split [59] in the experiments, which contains 113,287/5000/5000 training/validation/test images and each image is associated with 5 human-annotated captions.

**Implementation Details** We employ the Open-Flamingo model [16] to test our strategies, setting the length penalty to -2.0 and a maximum generation length of 20. We follow Flamingo [6] to use 4, 8, 16, and 32 shots. We respectively use ViT-L/14 and as the vision and language encoders to extract image and sentence embeddings that used in SIIR-CLIP and SICR-CLIP. For MGC-TF, we train the standard Transformer encoder-decoder architecture on the MSCOCO dataset and use the checkpoints underwent 1000, 3000, and 170,000 iterations respectively. These checkpoints generate the captions with CIDEr scores of 66, 88, and 135 on the test set. We implement all experiments on a single RTX 3090 using FP16.

### 4.2 Results and Analyses

Given our varied strategies for image selection and caption assignment, displaying results for each configuration in table format, especially at 4, 8, 16, and 32-shot levels, could become overwhelming. For clarity, we've chosen to present results using line charts and histograms in the main paper, while detailed numerical outcomes are in the supplementary material. The line charts in Figures 3 and 4 illustrate trends as the shot number grows. Each subplot within these figures corresponds to a unique strategy for image selection or caption assignment. Furthermore, Figures 5 and 6 show histograms of average CIDEr scores for the various shot results. To facilitate comprehension, we first analyze the effects of caption qualities 4.2.1 and then of image selection strategies 4.2.2.

### 4.2.1 Effects of Caption Qualities

Figure 4(a)-(e) reveals that performance typically improves with an increase in shot numbers. However, the rate of improvement varies, or even declines, depending on the quality of the captions used. For instance, in Figure 4(a), using Ground-Truth Captions (GTC), the increase rates of the

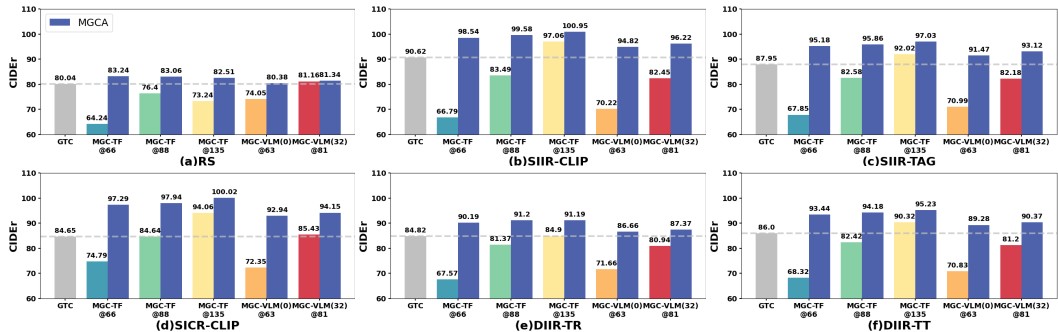

Figure 5: The histograms of various in-context captions with diverse image-selection strategies.

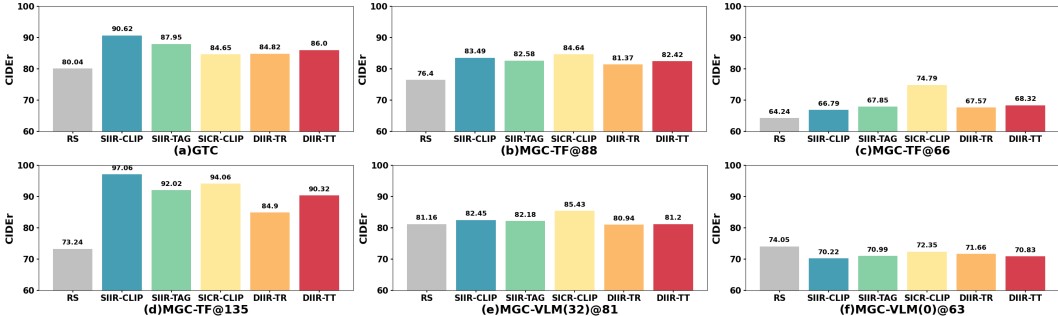

Figure 6: The histograms of various in-context images with diverse caption-assignment strategies.

six image selection strategies surpass those in Figure 4(b) where MGC-TF@88 is used. Further, low-quality captions, such as MGC-TF@66, may become "toxic examples" which can misguide the Vision-Language Model (VLM) as the shot number increases. Next we compare different kinds of captions to figure out what characteristics of the captions affect the performance of in-context captioning.

**Ground-Truth Captions (GTC) vs. Model-Generated Captions (MGC).** As discussed in Sec 3.2, MGC exhibits two primary shortcomings when compared to GTC: poorer language and less descriptiveness. However, we will see that these two shortcomings do not always make MGC achieve poorer performance when compared to GTC. Specifically, we find that *up to a certain level of descriptiveness, simpler sentence patterns are more easily recognized by the VLM, thereby improving caption generation.*

Evidence for this can be observed in Figure 5 by comparing GTC with MGC-TF@135. Compared to the caption generated by MGC-TF@135, ground-truth caption has better language patterns, *e.g.*, rich vocabulary and complex sentence pattern, and better descriptiveness. Then when the selected images cannot provide enough vision cues, *i.e.*, when Random Selection (RS) is applied in Figure 5 (a), GTC outperforms MGC-TF@135. However, once the similarity-based retrieval methods like Similarity-based Image-Image Retrieval (SIIR) or Similarity-based Image-Caption Retrieval (SICR-CLIP) is used, the selected similar images offer useful visual patterns that help address the descriptiveness issue. Consequently, the VLM is more likely to recognize the consistent, simple patterns in MGC than the rich, diverse patterns in GTC and then generate better captions. Figure 7 (a) visualizes 2 examples about the above comparisons.

This effect is more pronounced in 4-shot cases, where VLM lacks sufficient in-context captions to discern sentence patterns, making simpler patterns advantageous. As long as the descriptiveness issue is addressed, MGC often outperforms GTC, *e.g.*, MGC-TF@88>GTC in Figure 3(b-d). Especially, when SICR-CLIP is employed to select captions with high cross-modal similarity to the test image, it significantly mitigates the descriptiveness problem. Then as demonstrated in Figure 3(d), even MGC-TF@66 surpasses GTC.

**MGC-TF vs. MGC-VLM.** Before we see that using more low-quality captions like MGC-TF@66 will misguide VLM to generate worse captions. Yet, Figure 4 (f) indicates that using MGC-VLM(0)@63 improves performance with increased shot numbers, contrasting MGC-TF@66 in Figure 4 (c). This raises the question: why do weaker captions (MGC-VLM(0)@63) surpass those

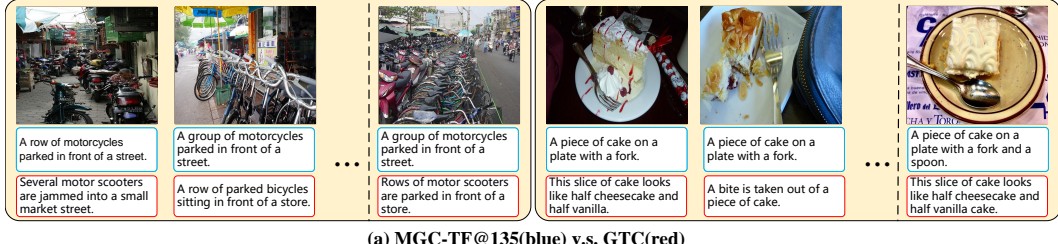

**(a) MGC-TF@135(blue) v.s. GTC(red)**

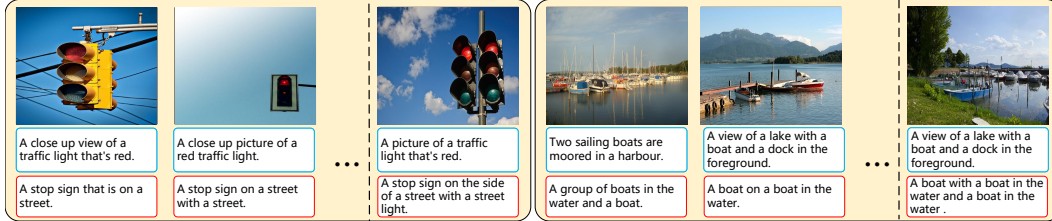

**(b) MGC-VLM(0)@63(blue) v.s. MGC-TF@66(red)**

Figure 7: (a) Two examples show that GTC uses more diverse words than MGC-TF@135, making VLM hard to recognize the major pattern, *e.g.*, it mis-generates "a store" in left or neglects "a fork" in right. (b) Two examples demonstrating how VLM is misguided by syntactic errors in MGC-VLM@66, where certain phrases are repeated such as "a street" or "a boat in the water".

with higher CIDEr (MGC-TF@66)? This discrepancy aligns with our assumption: descriptiveness and language pattern influence in-context captioning differently.

MGC-TF@66 excels in object detection but struggles with language decoding, causing salient objects to be identified correctly but with syntactical errors in captions. As such examples increase, VLM produces worse captions due to these errors. Conversely, MGC-VLM(0)@63, though limited in object recognition, maintains better grammar. When more vision cues are provided, VLM leverages these along with the better-formed captions from MGC-VLM(0)@63, resulting in improved captions. Figure 7 (b) offers two comparison examples.

**Model-Generated Captions as Anchors (MGCA).** From Figure 5, we see that using MGC as the in-context captions usually underperforms GTC. However, by the MGCA strategy, we observe consistent improvements over GTCs, as demonstrated by the higher blue histograms of different MGCs compared to the grey dashed line. For example, despite MGC-TF@66 only achieves 64.24 CIDEr score in Figure 5(a), we still observe a 3.2 CIDEr improvement when using MGC-TF@66 as the anchor, compared to simply selecting the first GTC. Specifically, when using MGC-TF@66/MGC-TF@88/MGC-TF@135/MGC-VLM(0)@63/MGC-VLM(32)@81 as the anchors, the average improvements over six image selection strategies compared to GTC are 7.3/8.0/8.8/3.6/4.8 respectively. In contrast to solely leveraging the RS+GTC method, the combination of SIIR-CLIP and using MGC-TF@135 as anchor chalked up an average boost of 20.9.

The primary reason for such improvement is likely that MGC, to some extent, verbalize the major patterns, such as the salient objects of an image. This helps identify which GTC provides more detailed information about these patterns. Such detailed information of the salient objects provide help VLM generate better captions. This assumption is further supported by comparisons between MGC-TF and MGC-VLM. Given that MGC-TF prioritizes verbalizing the salient objects of an image, using MGC-TF@66/MGC-TF@88 as anchors tends to select superior GTC than MGC-VLM(0)@63/MGC-VLM(32)@81, thus yielding higher improvements.

**Iteratively Prompting (IP).** Table 1 showcases the IP CIDEr scores. "Iter 1" represents either the 0 or 32-shot performance, with subsequent columns reflecting average CIDEr scores for 4, 8, 16, and 32-shot scenarios. In the inaugural iteration, we employ a consistent set of 0 or 32-shot image-caption pairs. For the following iterations, images are selected via the Random Sampling (RS) approach, and captions generated in the prior iteration serve as the in-context captions. Analyzing the data, it's evident that MGC-VLM(0) stabilizes by the third iteration and MGC-VLM(32) by

| Iter | 1 | 2 | 3 | 4 | 5 |
|---|---|---|---|---|---|
| MGC-VLM(0) | 63.0 | 74.1 | 79.9 | 79.3 | 77.3 |
| MGC-VLM(32) | 85.3 | 80.5 | 79.4 | 78.9 | 77.1 |

Table 1: The CIDEr scores of IP in different iterations.

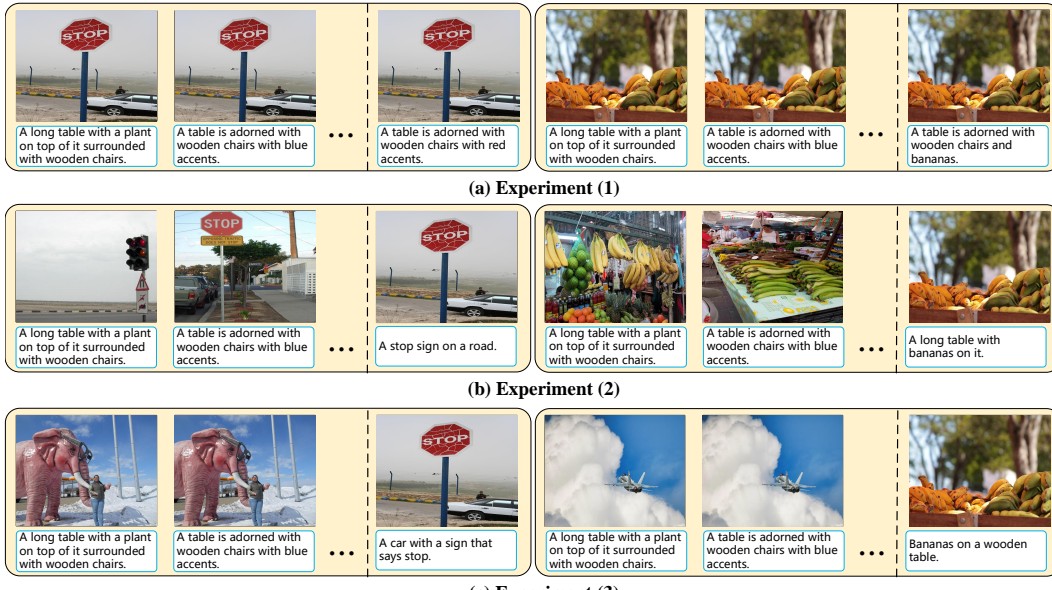

**(a) Experiment (1)**

**(b) Experiment (2)**

**(c) Experiment (3)**

Figure 8: Four examples illustrate the phenomenon of short-cut inference. (a) When the in-context images are identical to the test image, the generated caption mirrors the in-context captions. (b) When using SIIR-CLIP to select similar examples, the generated caption tends to amalgamate features from both the in-context and test images, sometimes leading to ambiguous or partially accurate descriptions. (c) In contrast, when the in-context images are distinct from the test image, the generated caption more aptly describes the image, including specific words such as "car" or "bananas".

the second, indicating that extended VLM iterations might be redundant. Remarkably, even when confined to just 32-shot GTCs, merely two iterations of IP — for instance, where MGC-VLM(32) attains an average CIDEr of 80.5 — can rival performances seen when all GTCs are utilized, as exemplified by RS-GTC's average CIDEr of 80.04 in Figure 5(a).

### 4.2.2 Effects of Image Qualities

We evaluated the outcomes of several image selection techniques. SIIR-CLIP, leveraging vision embeddings to compute retrieval similarities, generally identifies images that are more analogous than those found by SIIR-TAG. This is likely attributed to the intrinsic noise present in semantic tags. SICR-CLIP, emphasizing captions that spotlight prominent objects, naturally gravitates towards images showcasing similar objects. In contrast, both DIIR-TR and DIIR-TT produce more varied selections. Nevertheless, when benchmarked against RS, every retrieval-based model consistently fetches images that exhibit greater similarity.

At first glance, one could reasonably infer that using more similar images would invariably lead to superior performance. Yet, as demonstrated in Figure 6, this assumption doesn't universally hold. When engaging high-quality captions, namely GTC in (a) and MGC-TF@135 in (d), the correlation stands, with more analogous images indeed translating to enhanced results. For instance, all retrieval-based techniques surpass RS in this context. However, with medium-quality captions, as in MGC-TF@88 in (b) and VLM(32)@81 in (e), only the similarity-based retrieval methods like SIIR or SICR-CLIP manage to outdo RS, and this is specifically observed in (e). Most intriguingly, when low-quality captions like MGC-TF@66 in (c) and VLM(32)@63 in (f) are in play, analogous images inversely impact performance, as illustrated by RS outperforming SIIR-CLIP in (f). This critical insight underscores a pivotal revelation: *the efficacy of utilizing similar images is intricately tied to the caliber of the corresponding captions*.

**Similar Images Lead to Short-Cut Inference.** A relatively bold hypothesis to elucidate this phenomenon is: *when in-context images are similar to the test image, VLM may take a short-cut by leveraging in-context captions to generate a new one, rather than learning how to caption from the in-context pairs.* Consequently, the greater the similarity between in-context and test images, the more the VLM is influenced by the in-context captions.

To robustly test our underlying assumption, we designed three distinct experimental setups. In each setup, the in-context captions remain consistent, comprising 5 ground-truth captions sourced randomly from an image unrelated to the test image. However, the in-context images are chosen differently in each experiment: (1) they are identical to the test image; (2) they are picked using SIIR-CLIP; (3) they are chosen via RS. This progression results in decreasing similarity between the in-context and test images from experiments (1) through (3). Table 2 showcases two sets of CIDEr scores. One set compares the generated captions to the five ground-truth captions (GTC) of the test image, and the other set contrasts them with the five in-context captions (ICC). Our findings suggest a clear pattern: the closer the in-context images are to the test image, the more the VLM tends to mirror the ICC in its generated caption. For illustration, method (1) registers the highest CIDEr score when compared to the ICC. However, the caption it produces doesn't accurately depict the image, as reflected by its notably lower CIDEr score in relation to the GTC. These observations solidify our hypothesis that images with high similarity can inadvertently prompt short-cut inference. Additionally, Figure 8 provides visual representations to further elucidate the phenomenon of short-cut inference.

| | In-Context Images | GTC CIDEr | ICC CIDEr |
|---|---|---|---|
| (1) | Test Image | 6.7 | 192. 2 |
| (2) | SIIR-CLIP | 12.8 | 180.7 |
| (3) | RS | 58.5 | 54.8 |

Table 2: The results for verifying short-cut inference.

**DIIR.** As depicted in Figure 6, the performance of the two DIIR methods noticeably lags behind SIIR-TAG. This observation underscores the contention that the strength of diversity may not always translate to superior results in every context. One plausible explanation we propose is the inherent nature of captioning as a task. Contrary to certain complex NLP challenges, where diversity can be instrumental in offering a multifaceted understanding of a problem [57], captioning is relatively straightforward and may not benefit as much from diverse in-context examples. Delving deeper into the DIIR methods, DIIR-TT consistently outshines DIIR-TR. This leads us to infer that the clustering based on semantic tag types might be a more optimal strategy. Such a strategy, we believe, not only ensures diversity but also completeness in the selection of images. For instance, given a caption like "a brown dog is running", DIIR-TT could potentially source images that highlight elements like "brown objects", "dogs", and the "action of running".

## 5   Conclusion and Limitations

In this study, we utilize image captioning as a case study to examine the impacts of varied multi-modal in-context configurations on few-shot vision-language prompting. Specifically, we design 4 different ways to select images and 4 different strategies to assign captions to the selected images for constructing multi-modal in-context configurations. Exhaustive experiments reveal that, contrary to single-modal NLP cases, multi-modal mutual synergy significantly influences performance. Notably, we observe that the descriptiveness and language patterns of the captions differently affect performance: better performance may be achieved with simpler and consistent sentence patterns when selected images compensate for descriptiveness issues. And we also discover that when the in-context images are similar to the test one, VLM may build a short-cut by directly using the in-context captions instead of really learning to captioning. Moreover, our optimal strategy lead to a 20.9 average increase in CIDEr scores compared to a random sampling baseline.

This study's primary limitation is that at the inception of our exploration, the only open-source multi-modal few-shot learner available is Open-Flamingo [16]. However, Open-Flamingo, when compared with the official Flamingo [6], underperforms due to its training on significantly less data. Consequently, some findings in this paper might shift if a more robust multi-modal few-shot learner is employed. Nevertheless, even in such a scenario, the diverse configuration strategies proposed in this paper maintain their utility, aiding researchers in swiftly exploring the characteristics of the employed VLMs. Additionally, we have provided experimental results on Otter [54] and smaller version of Open-Flamingo, with detailed findings available in Appendix.

## Acknowledgments

This work is supported by National Science Foundation of China (62206048), Natural Science Foundation of Jiangsu Province (BK20220819), Young Elite Scientists Sponsorship Program of Jiangsu Association for Science and Technology Tj-2022-027 and the Big Data Computing Center of Southeast University.

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

# Supplementary Material

## A    Experimental results on Open-Flamingo v1 9B

Here we present all the numerical results from the experiment, divided into four tables (Table 3, Table 4, Table 5, and Table 6) based on different Image Selection strategies. Additionally, we have calculated the average values for each method in the "mean" column.

| Image Selection | Caption Assignment | 4-shot | 8-shot | 16-shot | 32 -shot | mean |
|---|---|---|---|---|---|---|
| RS | GTC | 75.80 | 78.93 | 81.97 | 83.47 | 80.04 |
| RS | MGC-TF@66 | 69.18 | 64.59 | 62.73 | 60.46 | 64.24 |
| RS | MGCA-TF@66 | 78.52 | 82.44 | 85.48 | 86.53 | 83.24 |
| RS | MGC-TF@88 | 74.60 | 74.47 | 77.80 | 78.71 | 76.39 |
| RS | MGCA-TF@88 | 78.40 | 81.28 | 84.93 | 87.62 | 83.06 |
| RS | MGC-TF@135 | 72.35 | 70.10 | 72.73 | 77.76 | 73.23 |
| RS | MGCA-TF@135 | 78.81 | 80.64 | 83.86 | 86.74 | 82.51 |
| RS | MGC-VLM(0)@63 | 70.45 | 73.92 | 74.83 | 77.00 | 74.05 |
| RS | MGCA-VLM(0)@63 | 76.13 | 79.61 | 82.14 | 83.63 | 80.38 |
| RS | MGC-VLM(32)@85 | 78.33 | 80.94 | 82.62 | 82.75 | 81.16 |
| RS | MGCA-VLM(32)@85 | 77.22 | 80.16 | 82.97 | 85.01 | 81.34 |

Table 3: Using Random Selection (RS) image selection strategy results.

| Image Selection | Caption Assignment | 4-shot | 8-shot | 16-shot | 32 -shot | mean |
|---|---|---|---|---|---|---|
| SIIR-CLIP | GTC | 80.32 | 88.76 | 95.18 | 98.24 | 90.62 |
| SIIR-CLIP | MGC-TF@66 | 69.08 | 66.78 | 65.79 | 65.51 | 66.79 |
| SIIR-CLIP | MGCA-TF@66 | 89.69 | 97.53 | 102.31 | 104.63 | 98.54 |
| SIIR-CLIP | MGC-TF@88 | 81.86 | 82.43 | 84.77 | 84.89 | 83.49 |
| SIIR-CLIP | MGCA-TF@88 | 91.11 | 98.23 | 102.74 | 106.24 | 99.58 |
| SIIR-CLIP | MGC-TF@135 | 95.64 | 96.62 | 97.66 | 98.32 | 97.06 |
| SIIR-CLIP | MGCA-TF@135 | 92.73 | 99.47 | 104.09 | 107.52 | 100.95 |
| SIIR-CLIP | MGC-VLM(0)@63 | 65.98 | 69.52 | 71.88 | 73.49 | 70.22 |
| SIIR-CLIP | MGCA-VLM(0)@63 | 86.72 | 93.30 | 98.42 | 100.85 | 94.82 |
| SIIR-CLIP | MGC-VLM(32)@85 | 79.46 | 81.34 | 84.14 | 84.86 | 82.45 |
| SIIR-CLIP | MGCA-VLM(32)@85 | 88.09 | 95.70 | 98.98 | 102.12 | 96.22 |
| SIIR-TAGS | GTC | 78.37 | 86.40 | 92.08 | 94.94 | 87.95 |
| SIIR-TAGS | MGC-TF@66 | 71.23 | 68.27 | 66.77 | 65.12 | 67.85 |
| SIIR-TAGS | MGCA-TF@66 | 87.40 | 93.31 | 97.85 | 102.16 | 95.18 |
| SIIR-TAGS | MGC-TF@88 | 82.70 | 81.80 | 82.51 | 83.33 | 82.58 |
| SIIR-TAGS | MGCA-TF@88 | 87.86 | 94.26 | 98.69 | 102.64 | 95.86 |
| SIIR-TAGS | MGC-TF@135 | 89.35 | 91.14 | 92.73 | 94.86 | 92.02 |
| SIIR-TAGS | MGCA-TF@135 | 88.89 | 94.45 | 100.34 | 104.44 | 97.03 |
| SIIR-TAGS | MGC-VLM(0)@63 | 67.11 | 70.21 | 72.21 | 74.45 | 71.00 |
| SIIR-TAGS | MGCA-VLM(0)@63 | 83.23 | 89.35 | 94.51 | 98.79 | 91.47 |
| SIIR-TAGS | MGC-VLM(32)@85 | 79.69 | 81.96 | 82.82 | 84.25 | 82.18 |
| SIIR-TAGS | MGCA-VLM(32)@85 | 85.04 | 91.23 | 96.10 | 100.10 | 93.12 |

Table 4: Using Similarity-based Image-Caption Retrieval (SIIR-CLIP) image selection strategy results.

| Image Selection | Caption Assignment | 4-shot | 8-shot | 16-shot | 32 -shot | mean |
|---|---|---|---|---|---|---|
| SICR-CLIP | GTC | 72.91 | 81.76 | 89.40 | 94.51 | 84.64 |
| SICR-CLIP | MGC-TF@66 | 76.03 | 74.66 | 74.73 | 73.75 | 74.79 |
| SICR-CLIP | MGCA-TF@66 | 87.50 | 95.67 | 101.56 | 104.42 | 97.29 |
| SICR-CLIP | MGC-TF@88 | 84.30 | 84.09 | 85.34 | 84.83 | 84.64 |
| SICR-CLIP | MGCA-TF@88 | 88.41 | 96.37 | 101.89 | 105.07 | 97.94 |
| SICR-CLIP | MGC-TF@135 | 92.60 | 93.69 | 94.64 | 95.30 | 94.06 |
| SICR-CLIP | MGCA-TF@135 | 91.19 | 98.41 | 103.24 | 107.24 | 100.02 |
| SICR-CLIP | MGC-VLM(0)@63 | 68.19 | 70.34 | 73.97 | 76.91 | 72.35 |
| SICR-CLIP | MGCA-VLM(0)@63 | 82.49 | 91.08 | 96.64 | 101.55 | 92.94 |
| SICR-CLIP | MGC-VLM(32)@85 | 82.39 | 85.02 | 87.22 | 87.08 | 85.43 |
| SICR-CLIP | MGCA-VLM(32)@85 | 84.18 | 91.52 | 98.02 | 102.89 | 94.15 |

Table 5: Using Similarity-based Image-Caption Retrieval (SICR) image selection strategy results.

| Image Selection | Caption Assignment | 4-shot | 8-shot | 16-shot | 32 -shot | mean |
|---|---|---|---|---|---|---|
| DIIR-TR | GTC | 76.01 | 82.88 | 87.13 | 93.25 | 84.82 |
| DIIR-TR | MGC-TF@66 | 71.45 | 66.82 | 66.22 | 65.80 | 67.57 |
| DIIR-TR | MGCA-TF@66 | 82.27 | 87.45 | 93.56 | 97.49 | 90.19 |
| DIIR-TR | MGC-TF@88 | 81.03 | 79.50 | 81.88 | 83.06 | 81.37 |
| DIIR-TR | MGCA-TF@88 | 82.85 | 88.84 | 93.93 | 99.18 | 91.20 |
| DIIR-TR | MGC-TF@135 | 83.59 | 82.63 | 84.33 | 89.03 | 84.89 |
| DIIR-TR | MGCA-TF@135 | 83.44 | 88.83 | 94.26 | 98.23 | 91.19 |
| DIIR-TR | MGC-VLM(0)@63 | 67.64 | 70.91 | 73.21 | 74.86 | 71.66 |
| DIIR-TR | MGCA-VLM(0)@63 | 78.68 | 84.41 | 89.96 | 93.59 | 86.66 |
| DIIR-TR | MGC-VLM(32)@85 | 77.35 | 80.72 | 81.97 | 83.72 | 80.94 |
| DIIR-TR | MGCA-VLM(32)@85 | 79.88 | 85.40 | 89.59 | 94.62 | 87.37 |
| DIIR-TT | GTC | 75.65 | 83.21 | 89.70 | 95.42 | 86.00 |
| DIIR-TT | MGC-TF@66 | 72.59 | 68.51 | 66.66 | 65.53 | 68.32 |
| DIIR-TT | MGCA-TF@66 | 84.88 | 91.36 | 96.81 | 100.73 | 93.44 |
| DIIR-TT | MGC-TF@88 | 81.94 | 81.76 | 82.19 | 83.80 | 82.42 |
| DIIR-TT | MGCA-TF@88 | 85.13 | 92.61 | 97.35 | 101.61 | 94.17 |
| DIIR-TT | MGC-TF@135 | 88.01 | 88.65 | 90.58 | 94.04 | 90.32 |
| DIIR-TT | MGCA-TF@135 | 86.77 | 93.25 | 97.92 | 102.97 | 95.23 |
| DIIR-TT | MGC-VLM(0)@63 | 66.42 | 69.64 | 72.71 | 74.56 | 70.83 |
| DIIR-TT | MGCA-VLM(0)@63 | 79.32 | 87.36 | 92.76 | 97.69 | 89.28 |
| DIIR-TT | MGC-VLM(32)@85 | 76.91 | 80.21 | 83.24 | 84.43 | 81.20 |
| DIIR-TT | MGCA-VLM(32)@85 | 81.39 | 87.83 | 93.64 | 98.61 | 90.37 |

Table 6: Using Diversity-based Image-Image Retrieval (DIIR) image selection strategy results.

# B   Experimental results on Open-Flamingo v2 3B

Here we present the numerical results on Open-Flamingo v2 3B model [3] in Table 7 based on two different Image Selection strategies. Additionally, we have calculated the average values for each method in the "mean" column.

From the values in the table, it can be seen that the trend is basically consistent with the v1 model. It can be considered that our strategy and analysis can be transferred to different models.

# C   Experimental results on Otter

Here we present the numerical results on Otter model in Table 8 based on two different Image Selection strategies. Additionally, we have calculated the average values for each method in the "mean" column.

---

[3]Project: `https://laion.ai/blog/open-flamingo-v2/`

| Image Selection | Caption Assignment | 4-shot | 8-shot | 16-shot | 32 -shot | mean |
|---|---|---|---|---|---|---|
| RS | GT | 77.90 | 86.14 | 90.80 | 93.17 | 88.47 |
| RS | MGC-TF@66 | 82.16 | 86.49 | 88.16 | 86.61 | 86.55 |
| RS | MGC-TF@88 | 82.47 | 88.43 | 91.81 | 93.83 | 90.12 |
| RS | MGC-TF@135 | 84.64 | 89.93 | 90.70 | 92.03 | 90.31 |
| RS | MGCA-TF@66 | 77.95 | 87.64 | 92.54 | 95.14 | 90.09 |
| RS | MGCA-TF@88 | 78.95 | 88.21 | 92.17 | 95.28 | 90.19 |
| RS | MGCA-TF@135 | 79.61 | 87.99 | 91.92 | 95.22 | 89.95 |
| SIIR | GT | 84.53 | 92.36 | 96.36 | 98.66 | 94.36 |
| SIIR | MGC-TF@66 | 84.72 | 77.69 | 75.29 | 76.49 | 77.09 |
| SIIR | MGC-TF@88 | 93.38 | 91.29 | 90.50 | 89.25 | 90.90 |
| SIIR | MGC-TF@135 | 104.10 | 103.80 | 103.31 | 102.39 | 103.56 |
| SIIR | MGCA-TF@66 | 90.00 | 97.51 | 102.04 | 103.50 | 99.78 |
| SIIR | MGCA-TF@88 | 91.15 | 98.88 | 102.17 | 104.55 | 100.53 |
| SIIR | MGCA-TF@135 | 91.91 | 99.01 | 103.26 | 104.93 | 101.14 |

Table 7: Open-Flamingo v2 3B results with various strategies.

From the values in the table, it can be seen that the trend is basically consistent with the v1 model. It can be considered that our strategy and analysis can be transferred to different models.

| Image Selection | Caption Assignment | 4-shot | 8-shot | 16-shot | 32 -shot | mean |
|---|---|---|---|---|---|---|
| RS | GT | 83.43 | 88.36 | 91.86 | 93.09 | 90.11 |
| RS | MGC-TF@135 | 75.49 | 80.77 | 85.46 | 89.53 | 83.115 |
| RS | MGCA-TF@135 | 84.67 | 89.6 | 92.45 | 93.97 | 91.025 |
| SIIR | GT | 87.4 | 90.72 | 94.42 | 95.94 | 92.57 |
| SIIR | MGC-TF@135 | 95.02 | 97.37 | 98.8 | 99.88 | 98.085 |
| SIIR | MGCA-TF@135 | 90.93 | 96.59 | 97.08 | 101.3 | 96.835 |

Table 8: Otter results with various strategies.

# D    More Results of MGC-TF@135 vs. GTC

We further elucidate the performance of MGC-TF@135(blue) and GTC(red), by offering additional examples shown in Figure 9. It can be easily observed that GTC has a more diverse range of sentence structures in comparison to MGC-TF@135. In the initial two examples, the words "cat" and "dachshunds" were inaccurately recognized by GTC, demonstrating its limitation in some specific instances. A shift can be noticed in the subsequent three examples, where the GTC generates captions with complex sentence structures, with a notable proportion of incorrect information. This finding serves to reinforce the notion that simpler sentence patterns, up to a certain degree of descriptiveness, are more readily deciphered by the VLM, thereby enhancing the quality of generated captions.

# E    More Results of MGC-TF@66 vs. MGC-VLM(0)@63

Similarly, we supplement with more examples, as depicted in Figure 10, to facilitate the comparison between MGC-TF@66(red) and MGC-VLM(0)@63(blue). A striking observation from this comparative study reveals that MGC-TF@66 demonstrates a significant number of syntactical errors. This flaw becomes problematic as it misdirects VLM, resulting in a substantial volume of syntactical mistakes in the produced sentences.This correlation implies that the syntax errors in the initial input by MGC-TF@66 tend to propagate into the VLM's output. Therefore, it becomes clear that the accuracy of grammar in the prompt is crucial for achieving better results.

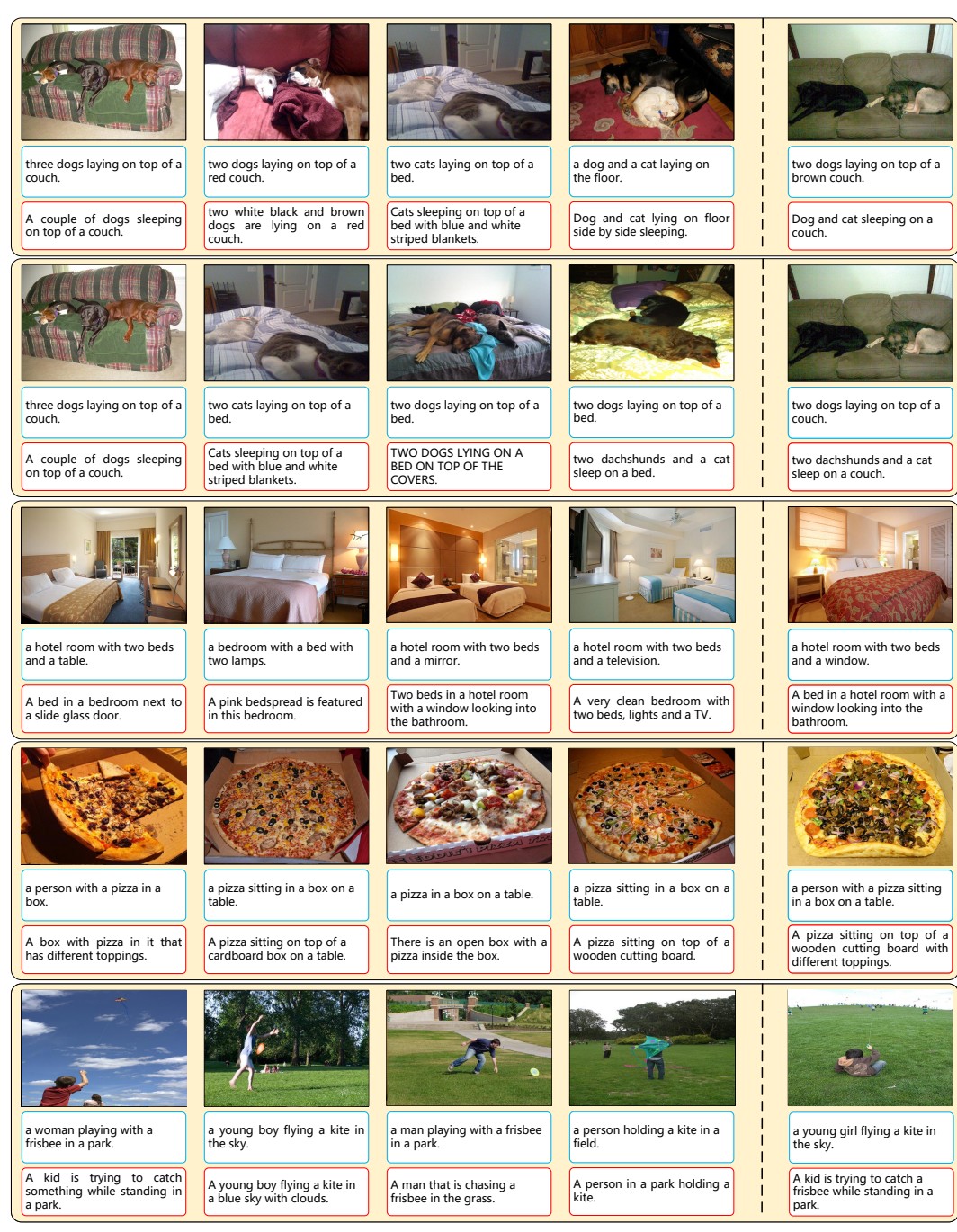

Figure 9: MGC-TF@135(blue) vs. GTC(red). Five examples demonstrate that more diverse words will be used in GTC than in MGC-TF@135 which making VLM hard to catch the major pattern. *e.g.*, the first line and the second line incorrectly state "cat" and "dachshunds."

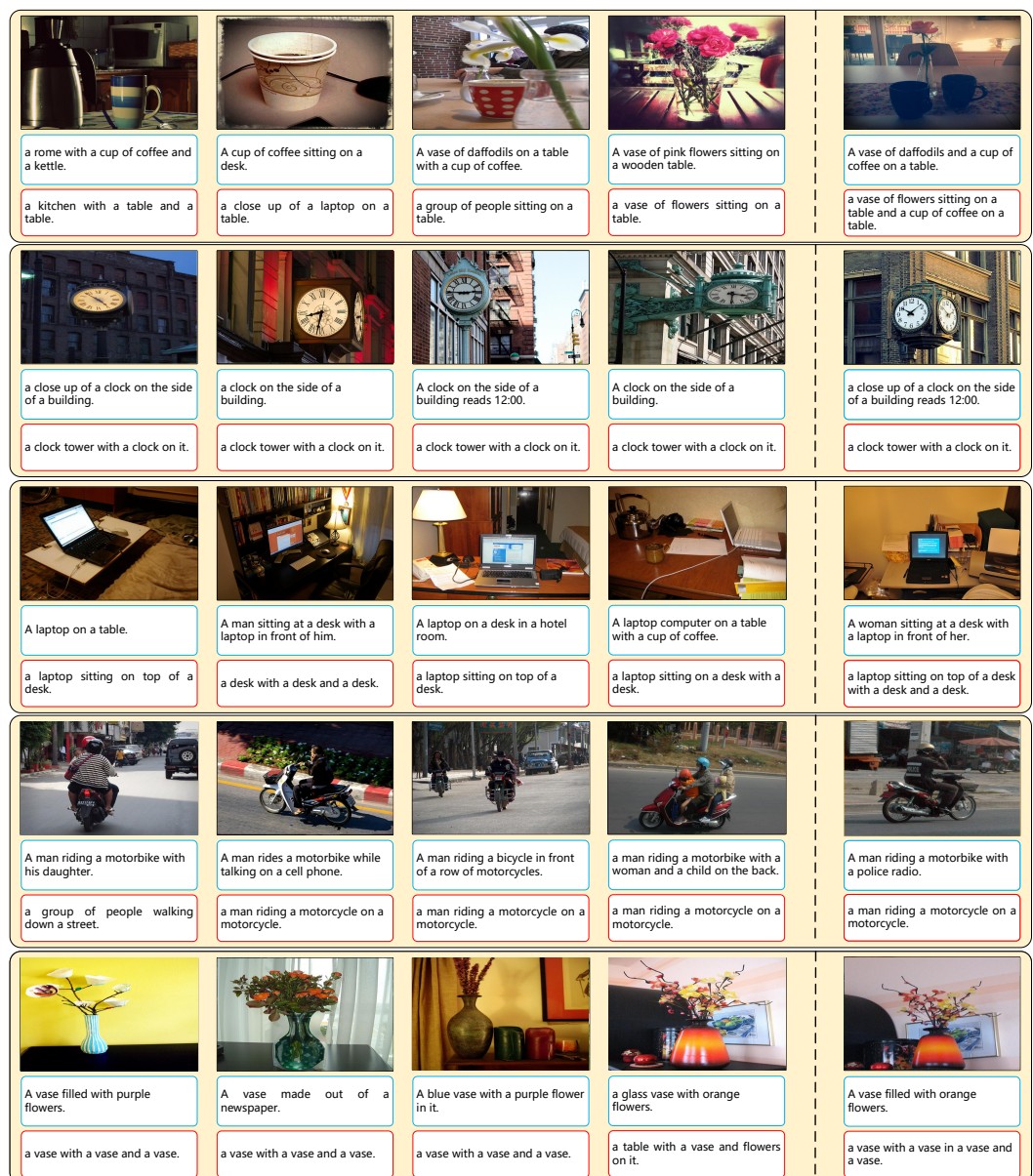

Figure 10: MGC-VLM(0)@63(blue) vs. MGC-TF@66(red). Five examples show that VLM will be misguided by the syntactic errors in MGC-VLM@66. Lots of outputs have repeated phrases, such as "a table", "a clock", "a desk", "a motorcycle" and "a vase".

# F   More Results of short-cut inference

In this section, we provide additional examples to demonstrate how similar images can lead our VLM to exhibit a tendency for short-cut inference, where it might relies heavily on in-context captions to generate the captions, disregarding the image information in the prompt and the inherent relationships within the in-context pairs. In Figure 11, we present the results for the same test image under two different scenarios of choosing in-context images: "identical to the test image" (top row) and "via random sampling" (bottom row).

In the first scenario, for the majority of cases, our results are largely unrelated to the image content but similar to the in-context captions, with only a few instances capturing some elements from the image, such as "pink" in (a) and "on the beach" in (b). However, in the second scenario, our results are heavily influenced by the in-context captions, as evident in (c) and (d) with the mention of "wooden chairs".

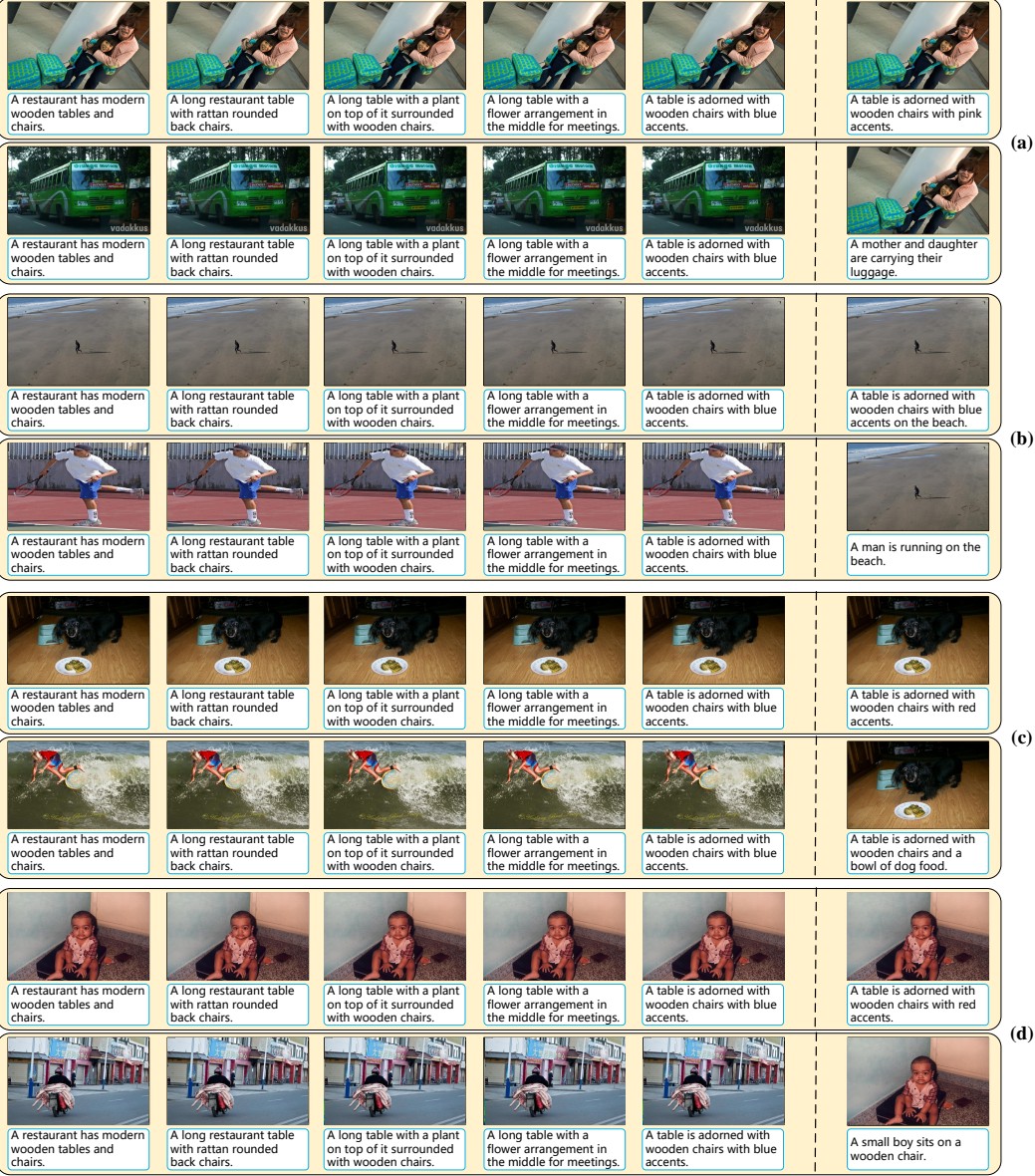

Figure 11: Five examples of verifying short-cut inference. we provide two different ways of choosing in-context images: "identical to the test image" (top row) and "via random sampling" (bottom row).

