# OpenReview forum: "Exploring Diverse In-Context Configurations for Image Captioning"
_NeurIPS.cc/2023/Conference — NeurIPS 2023 poster_

### Official Review · Reviewer_tg5h · 2023-07-03

**Soundness:** 3 good
**Presentation:** 2 fair
**Contribution:** 3 good
**Rating:** 6
**Confidence:** 4

**Summary:**

This paper presents a case study that examines the design of prompts and its effectiveness in generative visual language models for image captioning tasks. The authors delve into four methods for image selection and four methods for assigning captions to create contextually relevant learning samples. To evaluate different prompt design formats, the authors employed OpenFlamingo, an open-source version of the Flamingo paper. Several ablations and experiments were conducted to investigate the optimal inclusion of images and captions in the prompt of the Visual Language Model (VLM) to enhance performance. The authors put forth several findings:

1. Unlike single-modal NLP cases, the performance of the model is significantly influenced by the multi-modal mutual synergy.

2. The authors observe that the descriptiveness and language patterns of the captions have varying impacts on performance. In cases where selected images compensate for descriptiveness issues, better performance can be achieved by using simpler sentence patterns.

3. Additionally, the authors discover that when the in-context images resemble the test image, the VLM may take a shortcut by directly utilizing the in-context captions instead of genuinely learning to generate captions.

**Strengths:**

Generally, this paper is not novel, as there are similar papers in the NLP literature (see [1, 2]). While the authors mention [1] as one of the prior works, they appear to have overlooked [2].

In contrast to the aforementioned prior works, I believe this paper seeks to delve into prior studies in more complex tasks, specifically open-ended generation tasks such as image captioning, and employing visual language models. In terms of quality, the paper explores a distinct aspect of the research questions and presents compelling findings that align with some prior works. Experimentally, the paper demonstrates rigor by thoroughly exploring different prompt constructions.

[1]. "Rethinking the Role of Demonstrations: What Makes In-Context Learning Work?" EMNLP 2022.

[2]. "What Makes Good Examples for Visual In-Context Learning?" presented at ICML 2023.

**Weaknesses:**

The paper has a few notable weaknesses that can be addressed:

- The writing, particularly in lines 141-150, is challenging to comprehend. It would be beneficial for the authors to rephrase or provide visual aids to clarify this section, as it contains one of the paper's ablation studies.
- Sections 4.2.1 and 4.2.2 should be swapped, considering that in the pipeline, images are typically selected first, followed by the assignment of captions. The current ordering of the text makes it difficult to parse.
- It would be helpful to confirm the accuracy of the statement, "Then, as demonstrated in Figure 5(d), even MGC-TF@66 surpasses GTC."
- The author's choice of abbreviations throughout the manuscript makes the reading experience challenging. It requires readers to frequently refer back and forth between different parts of the text to fully understand a paragraph. It would be advisable to use less abbreviated phrasing.
- Including more elaborate captions for the figures would enhance their comprehensibility as standalone visual aids.
- The paper exclusively explores one visual language model, specifically OpenFlamingo, with 9B parameters. It is important to acknowledge that the findings may not be generalizable to other models.

**Questions:**

1. The paper could benefit from considering alternative visual language models to enhance the quality of the results. One such model is FROMAGe [1], which also utilizes a sequence of interleaved images and texts, similar to OpenFlamingo. It would be valuable for the authors to explore the use of FROMAGe in addition to OpenFlamingo and examine if the findings remain consistent across different models.

2. Additionally, it is worth investigating the impact of using smaller language models within OpenFlamingo. As per the OpenFlamingo GitHub repository, there are newer versions pre-trained on smaller LLMs. The authors could conduct ablations with these models to determine how the findings vary when using models with fewer parameters. This would provide insights into the robustness of the results across different model sizes.

[1]. Grounding Language Models to Images for Multimodal Inputs and Outputs, ICML 2023

**Limitations:**

There are not any limitations.

---

> ### Author Rebuttal · Authors · 2023-08-09
>
> **1. DIIR in Lines 141-150.**
> In Diversity-based Image-Image Retrieval (DIIR), we first extract the discrete semantic tags including object categories, attributes, and relations from an image. Suppose there are total $M$ kinds of tags, we divide them into $N$ clusters and there are $M/N$ kinds of tags in each cluster. Then in each tag cluster, we only use the tags in that cluster to calculate the similarity score between two images, i.e., the number of overlapped tags of two images. For DIIR-TR, it randomly divides all the tags into $N$ clusters, while DIIR-TT divides the tags according to their types, e.g., one cluster only contains attribute tags and another one only contains relation tags.
>
> **2. Section 4.2.1 and 4.2.2.**
> We will swap Section 4.2.1 and 4.2.2.
>
> **3. Accuracy Statement.**
> We will revise this statement into "Then, as demonstrated in Figure 5(d), in the 4-shot case, even MGC-TF@66 (the blue point) surpasses GTC (the grey point)."
>
> **4. Abbreviations.**
> We will use the full name at the beginning of each section and in the title of the tables and figures to avoid confusions.
>
> **5. Figure Titles.**
> We will add some key takeaways in some figures. Please refer to **"7. Takeaways."** in the response to the **Reviewer vuiF** to see two examples about Figure 3 and 4.
>
> **6. Experiments on FROMAGe.**
> Since FROMAGe does not provide codes for in-context learning (ICL) and lacks quantitative analyses in the paper, such as computing CIDEr scores, we modify parts of its code originally designed for multi-turn chat. We adopt a prompt structure like "<img1>caption1<img2>caption2...<test img>" for in-context learning. The test results are presented below.
> | Image Selection | Caption Assignment | 4-shot| 8-shot|
> |-----------------|--------------------|--------|--------|
> | RS              | GTC     | 2.50 | 3.11 |
>
> We can find that it achieves a very low CIDEr score.
> These observations are consistent with the analysis in the FROMAGe paper whose Figure.7 shows that when provided with 5-shot inputs, FROMAGe's output is more story-like and less relevant to the target image. Thus FROMAGe does not support the study about in-context configurations.
>
> **7. Experiments on Smaller Open-flamingo and Otter.**
> We test our methods in another two VLMs: [Open-Flamingo 3B model](https://github.com/mlfoundations/open_flamingo) and [Otter](https://github.com/Luodian/Otter), which are proposed **after the NeurIPS submission deadline**. However, due to the computational limitations and the tight rebuttal timeframe, we employ parts of experiments to validate the robustness of our two key conclusions shown in Lines 227-229 and 302-306.
>
> We first show the results about the first conclusion, where the top and bottom tables are from Open-Flamingo 3B model and Otter, respectively.
> | Image Selection | Caption Assignment | Mean CIDEr|
> |-----------------|--------------------|------|
> | RS              | GTC                | 90.31 |
> | RS              | MGC-TF@135         | 88.47|
> | SIIR-CLIP       | GTC                | 94.36 |
> | SIIR-CLIP       | MGC-TF@135         | 103.56|
>
> | Image Selection | Caption Assignment | Mean CIDEr|
> |-----------------|--------------------|------|
> | RS              | GTC                | 88.36 |
> | RS              | MGC-TF@135         | 80.77 |
> | SIIR-CLIP       | GTC                | 90.72 |
> | SIIR-CLIP       | MGC-TF@135         | 97.37 |
>
> where RS and SIIR-CLIP denote randomly sampling and clip Similarity-based Image-Image retrieval, GTC uses the first ground-truth caption among 5 human-labelled ones, and MGC-TF@135 uses the model-generated captions. We can find that in both two Tables, when RS is used to select images, GTC outperforms MGC-TF@135. For example, in the smaller open-flamingo model (the top one), RS-GTC achieves 90.31, which is higher than RS-MGC-TF@135's 88.47. However, when selecting similar images as in-context images, i.e., using SIIR-CLIP, MGC-TF@135 achieves higher performances, e.g., in the last two rows of the bottom table, MGC-TF@135 achieves 97.37, which is larger than GTC's 90.72. These comparisons in both two tables are consistent with the findings in Lines 230-238, which confirm the robustness of our conclusion given in Lines 227-229 that when the visual cues are abundant, the consistent and simple patterns of MGC has more advantages.
>
> Then we show the results about the second conclusion that similar images lead to short-cut inference, where the top and bottom ones show the results from the Open-Flamingo 3B model and Otter, respectively.
> |   | In-Context Images | GTC CIDEr | ICC  CIDEr |
> |---|-------------------|------------------|------------------|
> |(1)| Test Image        | 38.50             | 150.70            |
> |(2)| SIIR-CLIP         | 55.98             | 102.70            |
> |(3)| RS                | 70.72             | 43.50             |
>
> |   | In-Context Images | GTC CIDEr | ICC  CIDEr |
> |---|-------------------|------------------|------------------|
> |(1)| Test Image        | 66.40             | 37.90          |
> |(2)| SIIR-CLIP         | 73.04             | 27.90         |
> |(3)| RS                | 84.32             | 4.80           |
>
> From (1) to (3), we respectively use (1) the test image, (2) the retrieved similar images as the test image, (3) randomly sampled images as the in-context images, where the similarity to the test image from method (1) to (3) declines. Then we observe that in both tables, from (1) to (3), the CIDEr scores diverge more from in-context captions (ICC) but converge towards ground-truth captions (GTC). These results are consistent with the findings in Lines 315-319, and thus the conclusion "similar images lead to short-cut inference" still validates in smaller open-flamingo 3B model and Otter.
>
> **8. Limitation.**
> We acknowledge that the findings may not be generalizable to other models in Lines 340-346.

---

> > ### Comment · Reviewer_tg5h · 2023-08-20
> > **Post-rebuttal comments**
> >
> > Thanks authors for their detailed clarifications and for providing the experiment I requested. I have also reviewed the comments made by other reviewers and the authors' responses to them. The authors' rebuttal effectively addressed my concerns, and as a result, I will not be lowering my score.

---

> > > ### Author Response · Authors · 2023-08-20
> > >
> > > Thank you for taking the time to review our work and provide us with your feedback. We deeply appreciate your constructive comments and are pleased to hear that our clarifications and additional experiments were satisfactory. And we will further polish the writting by following your suggestions.

---

### Official Review · Reviewer_vuiF · 2023-07-07

**Soundness:** 3 good
**Presentation:** 1 poor
**Contribution:** 2 fair
**Rating:** 5
**Confidence:** 4

**Summary:**

The paper studies how to choose in-context examples (images and text) for open flamingo models on the CoCo captioning task. Extensive experiments are run with many baselines. Based on these experiments, the authors conclude (1) given sufficient descriptiveness, simpler captions are better, (2) similarity of in-context images to the test image is less important when the corresponding captions to said images are poor.

**Strengths:**

S1. Strong empirical investigation of how different image and text sampling strategies affect in-context image captioning performance. The paper is correctly framed and emphasizes experimentation over methodology.

S2. Extensive baselines implemented for the investigation.

S3. Seemingly strong performance relative to random sampling on the train set, which is traditionally used to generate in-context examples.

**Weaknesses:**

W1. Presentation. Consider adding some of the key findings to the abstract. 7.3 CIDEr points is significant! It may be nice to mention this in the abstract as well.

W2. Presentation. Some of the images in Figure 1 are hard to follow, consider making the example more demonstrative, or making the images bigger. Also grouping the good and bad captions within the same bubble was initially confusing and it took me some time to parse what was going on.

W3. Presentation. L51-59 potentially have too much detail for the introduction. Consider distilling the content. The key takeaway from this paragraphs seems to be that the authors experimented with many sampling strategies.

W4. Experiments. It would be nice to see similar analysis for the VQAv2 dataset, which is also supported in the OpenFlamingo evaluation suite. One natural question is do some of the findings for image captioning translate to VQA-like tasks, which are potentially more similar to the QA tasks shown in Figure 1.

W5. Presentation. Consider removing the abbreviations SIIR, DIIR, etc. from the intro. They were a bit tough for me to follow as they are not standard to the best of my knowledge and made reading the intro more difficult. In general, the many abbreviations in the paper make the writing difficult to follow.

W6. Related work. Consider adding a related work section on image captioning, since this is a focus of the paper.

W7. Clarity. A bit of a nit, but please specify how how the samples are randomly sampled. I am assuming uniformly at random?

W8. Clarity. Can you please expand on the SIIR-TAG baseline? In DIIR, how can SIIR-TAG be applied to a collection of images (a cluster)?

W9. Missing baseline. What about using models like BLIP2 for synthetic captioning (MGC)?

W10. Clairity. It is hard to tell what the takeaways from Figures 3 and 4 are from glancing at them, especially because all of the abbreviations can be confusing. Consider adding they key takeaways to the figure captions. It is not currently clear to me how to read these figures. Is the takeaway that MGC is best? In general, it is not always clear how the conclusions or claims in the paper are supported numerically by the figures or tables (e.g., claims in L231-233).

W11. Presentation. In some cases numbers are averaged over multiple shots and in other cases this is not the case. Why did the authors choose to do this? And why is there not consistency in this choice?

**Questions:**

The following questions are distilled from my most significant concerns:

Can the authors address the concerns surrounding presentation (W1, W2, W3, W5, W7, W8, W10, W11)? While the baselines seem thorough, they are not always clear for me to follow. Additionally, while the authors run many experiments, the presentation of these experiments is not always interpretable and it is not clear to me that the experiments actually support the conclusions that the authors draw. Some more clarity surrounding this would be greatly appreciated.

Is it possible to do some experiments for the best, middle, and worst baselines on VQAv2 to see if trends are similar for another image to text task (W4)?

**Limitations:**

The authors acknowledge that findings for the open flamingo model tested may not translate to the more powerful, but also closed-source flamingo model.

---

> ### Author Rebuttal · Authors · 2023-08-09
>
> **1. Presentation Issues.**
> We will add key findings in "Abstract" (W1), enlarge Figure 1 (W2), divide the good and bad captions in Figure 1 (W2), distill the contents in Lines 51-59 (W3),  and use the full names of different sampling strategies in "Introduction" and at the beginning of each sections to avoid confusions (W5).
>
> **2. VQA Results (W4).**
> Please refer to **"3. VQA Results."** in response to **Reviewer zeNE**.
>
> **3. Related Work about Captioning (W6).**
> Due to the space limitation, we show a short version of the related work about captioning and will expand it in the revision.
>
> Image Captioning (IC)[A] aims at correctly verbalizing one image by descriptive languages, which can be solved through both retrieve[B] and generation[A], where the former one seeks a whole sentence from a given corpus, while the latter one generates the words one by one. Researchers have recently combined these approaches by first retrieving image-caption pairs and inputting them into generation models[C,D]. This process resembles in-context captioning, which also retrieves image-caption pairs to help captioning. However, in contrast to them[C,D], our work introduces novel image and caption selection strategies to study in-context captioning and these methods can also enhance traditional retrieval-generation methods.
>
> [A] Show and tell: A neural image caption generator.
>
> [B] Probabilistic Embeddings for Cross-Modal Retrieval.
>
> [C] SmallCap: Lightweight Image Captioning Prompted with Retrieval Augmentation.
>
> [D] Retrieval-augmented transformer for image captioning.
>
> **4. Randomly Sampling (W7).**
> "Randomly Sampling" means uniformly sampling and we will revise it.
>
> **5. SIIR-TAG (W8).**
> SIIR-TAG first calculates how many objects, attributes, and relations tags are overlapped between the images with the test image and then returns the ones which have more overlapping tags.
> For example, given one test image containing three tags "dog", "white", "drink", the image A contains three tags "cat", "white", "sit", the image B contains three tags "dog", "brown", "drink". Then SIIR-TAG will return image B since it has two overlapping tags with the test one. To efficiently count how many tags are overlapping, we list all the discrete tags in a one-hot manner and then apply the "AND" operation.
>
> Please refer to **"1. DIIR in Lines141-150."** in the response to **Reviewer tg5h** to see details of Diversity-based Image-Image Retrieval (DIIR). Briefly, DIIR divides all the tags into different clusters. Then when calculating the ratio of the overlapped tags, only the tags in that cluster are used. This is how we use SIIR-TAG to retrieve images in each cluster.
>
> **6. BLIP2 Captions (W9).**
> We use the official BLIP2 code from [link](https://github.com/salesforce/LAVIS/tree/main/projects/blip2) to produce model-generated captions (MGC) with a CIDEr score of 124. The subsequent table presents results using Random Sampling (RS) and CLIP Similarity-based Image-Image Retrieval (SIIR-CLIP) for image selection.
>
> | Image Selection | Caption Assignment | Mean CIDEr|
> |-----------------|--------------------|--------|
> | RS              | MGC-BLIP2@124      | 71.76 |
> | RS              | MGC-TF@88          | 76.40 |
> | SIIR-CLIP       | MGC-BLIP2@124      | 89.67 |
> | SIIR-CLIP       | MGC-TF@88          | 83.49 |
>
> From the table, despite BLIP2 captions achieving a CIDEr score of 124, outperforming MGC-TF@88's score of 88, it only surpasses MGC-TF@88 in the SIIR-CLIP case. This might be because BLIP2, trained on various captioning datasets, introduces styles different from COCO. When randomly sampling some images, this diversity might affect VLM's caption quality. However, BLIP2's detailed descriptions, due to its extensive training, enhance VLM's outputs when in-context images are similar to the test image. This highlights the interdependency of image and caption selection strategies.
>
> **7. Takeaways (W10).**
> We will highlight the key takeaways in figure titles. For instance, Figure 3 suggests that regardless of the image-selection strategy employed, the use of MGCA (using model-generated captions to select a ground-truth caption as the in-context caption) consistently outperforms GTC (simply selecting the first ground-truth caption as the in-context caption). This is evident as the blue blocks surpass the grey dash lines.
>
> In Figure 4, each sub-figure shows captions of varying quality. Across these sub-figures, six image selection strategies yield diverse rankings, e.g. SIIR-CLIP tops in (a) and (d) but performs less optimally in other sub-figures. A primary insight here is that the image selection strategy must adapt when using captions of diverse quality to ensure optimal performance.
>
> **8. Claim in Lines 231-233 (W10).**
> The claim in Lines 231-233 (ground-truth caption has better language patterns than the model-generated captions) is not a claim observed from our experiments, but is a consensus in image captioning. We employ this consensus as a foundation for analyzing our experimental results to support our claim in Lines 227-229. We will add one footnote in the revision to avoid confusion.
>
> **9. Average or Single-Shot Scores (W11).**
> During analysis, in most cases, as the number of shots changes, we observe consistent conclusions, leading us to utilize average scores. For example, in Lines 259-273, MGCA consistently improves performance irrespective of the shot number.  However, in some cases, the shot count becomes pivotal, revealing interesting details. For example, as highlighted in Lines 239-240, when only scarce 4-shot in-context examples are available, simpler patterns can be better recognized by VLM for better captions, as shown in Figure 5 (b-d) where MGC-TF@88 > GTC in the 4-shot scenario. But as shot numbers increase to 16-shots, VLM can better recognize the complex patterns of GTCs, producing superior results as illustrated in Figure 5 (b-d) where GTC > MGC-TF@88 in the 16-shot scenario.

---

> > ### Comment · Reviewer_vuiF · 2023-08-14
> >
> > Thanks to the authors for their detailed responses and additional experiments! I especially appreciate the VQAv2 results and BLIP2 captioning results. My outstanding complaint is the paper writing, which is I find to be inaccessible, especially with all the abbreviations. I find that the key takeaways are not always clear. An alternative approach would be to provide a table with the axes of variation in methodology that you all consider. You can then provide checkmarks do differentiate the different methods. I am electing to raise my score to a 5.

---

> > > ### Author Response · Authors · 2023-08-15
> > >
> > > Thanks for your appreciation of our detailed response. We will further polish the writting by following your suggestions.

---

### Official Review · Reviewer_Qg6d · 2023-07-07

**Soundness:** 4 excellent
**Presentation:** 3 good
**Contribution:** 3 good
**Rating:** 5
**Confidence:** 4

**Summary:**

This paper examines the effects of varying configurations on Vision-Language (VL) in-context learning, specifically in the field of image captioning. The authors developed four strategies each for image selection and caption assignment to determine how different methods of configuring image-text pairs impact the learning process. The study revealed two insights: 1) Caption quality, influenced by descriptiveness and language patterns, impacts captioning performance, with simpler language yielding better results when paired with descriptive captions. 2) The effectiveness of similar images relies on caption quality, with excessive similarity potentially misleading the model when paired with low-quality captions.

**Strengths:**

+ The general goal is reasonable and interesting. The field of in-context learning in multimodality is well worth exploring.

+ The experiments were comprehensive. The authors compared numerous in-context learning strategies, including some new proposals of their own.

+ The paper provides some useful insights about how to choose in-context samples.

**Weaknesses:**

- The authors said that “the performance ... heavily relies on the caption quality” in Fig 1 Caption. I'm curious about how the authors define the quality of a caption. The example given in Figure 1 suggests that the caption in the blue box is better than the one in the red box. It is easy to understand because the red box caption contains clear errors, such as repeated phrases. When selecting in-context samples, such samples obviously should not be selected. But, in practice, such samples should not even be available as candidates, right? Could the authors display what captions the model considers as high-quality or low-quality? Or intuitively illustrate what captions are expected to consider as a high quality among several ground truths?

- One insight of the paper is that "When captions adequately describe salient image objects, simpler language patterns may yield better
results." But aren't the ground truth captions in the training set in the language style expected by the benchmark? If we use captions generated by the model, then how does the model know what style is required by the benchmark?

- I'm not certain whether the experimental setting in this paper can effectively evaluate the capabilities of in-context learning. In-context learning aims to use a small number of samples as demonstrations to learn some new things, such as a new task, new input-output format,  new object naming rules, etc. This work conducts experiments on the MSCOCO caption dataset. I'm unsure what the purpose of in-context learning is in this case, as these samples seem quite common for large models. So, what patterns or things are expected to learn by model from these in-context samples?

**Questions:**

My major concerns are about the current setting of in-context learning, and I would like to hear more about how to define the quality of the captions, which is important in choosing samples.

---

> ### Author Rebuttal · Authors · 2023-08-06
>
> **1. Quality of Image Captions.**
> Image captioning quality hinges on the ability of the caption  to accurately and grammatically describe the main events of an image. Inadequate grammar or descriptions that either neglect or misinterpret key events indicate poor quality. Human-labeled ground truth captions (GTC) typically adhere to these standards and are thus deemed high-quality. This paper assesses various model-generated captions (MGC) with adjustable quality to explore their influence on in-context captioning. Directly quantifying MGC quality based on the aforementioned standards is challenging. Hence, researchers suggest computing the similarity between MGC and several human-labeled captions using CIDEr[A] as the primary metric. As noted in lines 167-172, we allocate a CIDEr score to each MGC to measure its quality, e.g., MGC-TF@66 implies the model-generated captions with a CIDEr score of 66. Higher CIDEr scores generally suggest superior quality. Our experiments reveal that varying caption qualities, denoted by different CIDEr scores, impact the final outcomes.
>
> Regarding Figure 1, it underscores that even when the used in-context images are similar to the test one, if the in-context captions have low-quality, the generated captions will still have low quality. To emphasize this, we present examples of particularly low-quality captions.
>
> **2. Diversity in Ground-Truth Caption Patterns.**
> Since ground-truth captions (GTC) are labelled by different annotators, they usually have much more diverse sentence formats compared to model-generated captions (MGC). Consider the diverse styles of these four GTC from the COCO dataset which are used as in-context captions:
> - A homemade pizza with sauce and cheese on foil.
> - Man leaning his mouth down to a plate that has a sandwich on it and a blue water bottle behind it.
> - Bunches of bananas hanging from fire on a line.
> - There are several birds that are standing at the beach.
>
> Then as demonstrated in Lines 236-237, we show that such diverse patterns can be more challenging for a VLM to recognize compared to simple, uniform patterns, hence MGC sometimes outperforms GTC.
>
> For the Transformer models shown in Lines 202-205 that are used to generate MGC, they are trained following [B] where the CIDEr scores between the generated captions with ground-truth captions are used as the objectives. In this way, these Transformer models will still learn the sentence style of the GTC captions.
>
> For more discussions about caption quality and patterns, you can refer to **"1. Consistent Format."** and **"2. Salient Objects."** in the response to the **Reviewer FMu1**.
>
> **3. Significance of Our Study.**
> We agree with you that the primary objective of In-context learning (ICL) is to use a few samples to learn new knowledge. However, this is not the major motivation for our research. Instead, we aim to explore how different image and caption selection strategies affect the performance of multi-modal ICL. The significance of our research is that it provides a roadmap for researchers in multi-modal ICL. For instance, our research shows that, unlike the NLP domain where using similar samples as in-context examples usually leads to good performance, in the multi-modal domain, the strategies for selecting images and texts influence each other, e.g., one major finding of our study is that it is not guaranteed that choosing similar images and high-quality texts will always yield the best results.
>
> [A] Cider: Consensus-based image description evaluation.
>
> [B] Self-critical Sequence Training for Image Captioning

---

> > ### Comment · Reviewer_Qg6d · 2023-08-18
> >
> > Thank you for the author's detailed response and the comments from other peers. The author's answers have addressed my main concerns, and I will raise my score.

---

> > > ### Author Response · Authors · 2023-08-19
> > >
> > > Thank you for reconsidering your score and acknowledging our detailed response. We appreciate your constructive feedback throughout this review process.

---

### Official Review · Reviewer_FMu1 · 2023-07-10

**Soundness:** 3 good
**Presentation:** 3 good
**Contribution:** 3 good
**Rating:** 5
**Confidence:** 3

**Summary:**

This work investigates the configuration of in-context image-caption pairs for few-shot image captioning in a VL model: OpenFlamingo. Specifically, authors compare four different image selection methods: (1) random selection, (2) image-image similarity based selection (SIIR), (3) image-caption similarity based selection (SICR), and (4) image-image diversity based selection (DIIR).

On top of the selected images, the authors also test out four different caption assignment methods: (1) ground-truth caption (GTC), (2) model-generated caption (MGC), (3) Iterative prompting (IP), and (4) Model-generated Captions as Anchors (MGCA).

The findings from this work suggest that (1) selecting similar images with the test image is better than random sampling; (2) simpler language patterns in captions may be better if they adequately explain the salient info of the image; and (3)similar in-context images can induce short-cut inference from their captions.

**Strengths:**

Analyses on in-context sampling method for VL models can expand the literature. Especially, the fact that MGCA results lead to better performance than random ground-truth captions is interesting.

**Weaknesses:**

All experiments are conducted on a single open-source VL model; this leads to questions whether these results hold generalizability.

Further analyses should have been conducted to backup the authors’ claims: “simpler caption pattern improves caption generation” & “verbalizing the major patterns help identify which GTC provides more detailed info about these patterns”. Please see the question section.

**Questions:**

(1) I find it interesting that GTC performs worse most of the time compared to MGC. Why do you think this the case? I am very suspicious that it is due to the consistent format of captions across in-context samples instead of the simple pattern of the MGCs. However, this is hard to analyze with the current setup. MGCs will have very similar sentence structures for similar images compared to GTCs, which is also shown in Figure 7-(a). I suggest controlling the format consistency across GTCs within the in-context samples. For example, you can set captions to have the same complex structure and see how that results in performance. If simplicity is the true cause of improved performance, setting consistent but complex format should not give improve performance. Also, you can try very diverse format across captions but make them simple in structure.

(2) Don’t almost all GTCs include the most salient object in their captions? I find it hard to believe that MGCA improves performance because it finds the GTC that describes the salient pattern with more info. The selected GTCs from MGCA will probably have similar structure with the MGCs, hence it reduces the format variance of GTCs across the in-context samples. What happens if you select the GTC that is least similar with the MGC?

(3) When giving the in-context samples, do the images really help in performance? What if you drop all in-context images and give only the captions instead?

(4) I think it would be better to swap the location of Figure 3 & 4 with Figure 5 & 6. Is there a reason why you put Figure 3 & 4 first? I found it difficult to go back and forth because the main draft explains Figure 5 & 6 first.

**Limitations:**

Yes, the authors also acknowledge the limitation that these analyses were conducted on a single model.

---

> ### Author Rebuttal · Authors · 2023-08-09
>
> **1. Consistent Format.**
> When we say that the model-generated captions (MGC) have "simple patterns", part of the meaning we wish to convey is that they often have a consistent format for captions, e.g., in Lines 236-237, we mention that ground-truth captions (GTC) contain more diverse patterns than MGC, which challenges the VLM to recognize one uniform format. We agree with you that the statement of "consistent format of captions" is more accurate than "simple pattern" and we will use this statement in the revision.
>
> To further validate this claim and follow your suggestion, we conduct an additional experiment by manually selecting 1000 captions which have a consistent format and building a candidate set $\mathcal{D}$ by these selected captions and their corresponding images. Then we apply random sampling (RS) to select images and use these captions with a more consistent format as in-context captions to build a baseline "CF" shown in the following table.
> | Image Selection | Caption Assignment | Mean CIDEr|
> |-----------------|--------------------|--------|
> | RS | GTC | 80.04 |
> | RS | CF |82.25 |
> | RS | MGCA-TF@88 | 83.06 |
>
> Note that "GTC" used in this table denotes to use the first ground-truth caption among 5 human-labelled ones and MGCA-TF@88 denotes to select one ground-truth caption by using the caption of MGC-TF@88 as the anchor. It can be seen that using the captions with a consistent format outperforms GTC, suggesting that consistent format plays a significant role in in-context captioning. However, MGCA still outperforms CF, suggesting that consistent format is not the only factor that improves the performance.
>
> **2. Salient Objects.**
> One image usually contains lots of objects where some are salient and some are not. However, different human annotators have diverse preferences in that they may describe different details of an image, including some non-salient objects. For example, [one image](https://cocodataset.org/#explore?id=280810) from the COCO dataset has the following ground-truth captions:
> - a small plate holds a pot pie and broccoli.
> - pot pie and broccoli on a plate on a table with a laptop in the background
> - a plate holds a meat pie, carrots and broccoli.
> - lunch is in front of the laptop computer.
> - a red and white plate topped with a pot pie and broccoli.
>
> We can find that only the 2nd one contains both the objects "plate" and "laptop". Then an MGC "a plate near a laptop" helps select the 2nd caption, which also remedies some details about the plate that may help generate better captions as discussed in Lines 268-270.
>
> To quantitatively verify the conclusion, we use VinVL [A] to extract the salient objects from each image based on the predicted probabilities and then calculate how many salient objects are overlapped with the nouns in the selected caption. We use three kinds of captions, (1) the caption which has the highest CIDEr score with the MGC, which gets **0.954** overlapping ratio, (2) the first caption among five ground-truth captions, which gets **0.845** overlapping ratio, and (3) the caption which has the lowest CIDEr score with the MGC, which gets **0.737** overlapping ratio. These findings confirm that MGCA-selected captions are more likely to contain salient objects, leading to improved performance.
>
> We also conduct an experiment using the caption with the lowest score to MGCA as our in-context captions. Some results are given here:
> | Image Selection | Anchor | MGCA Score | Mean CIDEr|
> |-----------------|----------------|---------|------|
> | RS              | GTC            | --      | 80.45 |
> | RS              | MGC-TF@88      | highest | 83.11 |
> | RS              | MGC-TF@88      | lowest  | 73.84 |
> | SIIR-CLIP       | GTC            | --      | 91.97 |
> | SIIR-CLIP       | MGC-TF@88      | highest | 99.58 |
> | SIIR-CLIP       | MGC-TF@88      | lowest  | 74.39 |
>
> We see that selecting the caption with the lowest score to MGCA greatly reduces performance, suggesting that the captions with the lowest scores contain fewer salient objects.
>
>
> **3. Dropping In-Context Images.**
> We follow your suggestion to drop all in-context images and give only the captions in in-context samples. The results are given in the following, where the RS and GTC strategies are used to select the images and captions:
> |      | Mean CIDEr|
> |-----------|-------|
> | $w$ image   | 80.04 |
> | $w/o$ image | 2.99  |
>
> We can find that without the images, the model cannot generate meaningful captions. We check the generated captions and find that they usually trend to repeat some of the provided in-context captions, suggesting the importance of the visual cues of the in-context images.
>
>
> **4. Figure 3-6.**
> We will follow your suggestion to swap Figure 3 & 4 with Figure 5 & 6.
>
> **5. Generalizability of the Key Conclusions.**
> Please refer to **"7.Experiments on Smaller Open-flamingo and Otter."** in the response to the **Reviewer tg5h**.
>
> [A] VinVL: Revisiting Visual Representations in Vision-Language Models

---

> > ### Comment · Reviewer_FMu1 · 2023-08-11
> >
> > I have read the authors' response. Thank you for running the additional experiments. It would be great to see these experiments included in the updated version.
> >
> > p.s.
> > I should've suggested randomly swapping the images rather than dropping all the images, as that would lead to a serious OOD and performance crash.

---

> > > ### Author Response · Authors · 2023-08-11
> > > **What ''randomly swapping the images'' means?**
> > >
> > > Thanks for your appreciation of our response. What ''randomly swapping the images'' means, should we swap the order of the in-context image-caption pairs or should we swap the images to make mismatched image-caption pairs?

---

> > > > ### Comment · Reviewer_FMu1 · 2023-08-12
> > > >
> > > > Sorry for the ambiguity, I was referring to the mismatched image-caption pairs.

---

> > > > > ### Author Response · Authors · 2023-08-12
> > > > > **The results of swapping the images**
> > > > >
> > > > > We follow your suggestion to swap image locations to produce mismatched pairs. The results are presented in the table below. We use Random sampling (RS) and CLIP embedding Similarity-based Image-Image Retrieval (SIIR-CLIP) as our image selection strategy, and take the first ground-truth caption as caption assignment. "Default" indicates that we did not change the image positions, while "Swap" signifies that we randomly shuffled the image locations. We find that, across both image selection strategies, "Swap" resulted in a performance drop. The decline is more pronounced in RS while for SIIR-CLIP is relatively slight, likely because it utilizes images that are similar as in-context examples, thus the mismatched captions don't introduce as much confusion.
> > > > >
> > > > > In summary, mismatches would lead to a negative impact. Since SIIR-CLIP is not a strict mismatch, its effect is minimal.
> > > > >
> > > > > | Image Selection | Image Location | Mean CIDEr |
> > > > > |-----------------|--------------------|------------|
> > > > > | RS              | Default            | 80.45      |
> > > > > | RS              | Swap               | 76.65      |
> > > > > | SIIR-CLIP       | Default            | 91.97      |
> > > > > | SIIR-CLIP       | Swap               | 91.27     |
> > > > >
> > > > > We hope this addresses your question.

---

> > > > > > ### Comment · Reviewer_FMu1 · 2023-08-12
> > > > > >
> > > > > > I didn't expect the additional experiment but really appreciate the extra effort. Thank you!
> > > > > > What is the number of samples for this experiment?

---

> > > > > > > ### Author Response · Authors · 2023-08-12
> > > > > > >
> > > > > > > We use the Karpathy split, resulting in 5,000 test samples. The results are the average values for 4/8/16/32-shot. We run the experiment because it is straightforward, and we are also curious about the potential outcomes. Thank you very much for your suggestion.

---

### Official Review · Reviewer_zeNe · 2023-07-31

**Soundness:** 2 fair
**Presentation:** 2 fair
**Contribution:** 3 good
**Rating:** 5
**Confidence:** 5

**Summary:**

This paper explores the in-context configurations for few-shot ability in Vision-Language Models (VLMs). Therefore, they design four strategies for image selection and four for caption assignment to explore the influences of in-context pairs in image captioning. As a result, extensive experiments uncover two valuable insights (1) The captions adequately describe salient image objects and simpler language patterns may yield better results. (2) Excessive similarity might cause VLMs to create a short-cut inference from in-context captions. Finally, they introduce the iterative prompting for images with limited or no ground-truth captions, which boosts the performance of models with an average CIDEr improvement of 7.3.

**Strengths:**

1.	**The considered problem is very relevant and timely for the AI community.**. It is urgent to explore the in-context configurations for vision-language pre-trained models.
2.	**The observations are valuable.**. They observe that better performance may be achieved with simpler sentence patterns when selected images compensate for descriptiveness issues. Moreover, they observe that the VLM may build a shortcut rather than learn to caption when the in-context images are similar to the test one.
3.    **The experiments are comprehensive.**. They devised four strategies for image selection and four for caption assignment to explore the influences of in-context pairs in image captioning.


**Weaknesses:**

1.	**The selected metric is not convinced.** They use CIDEr to evaluate caption models' performances. However, CIDEr is a statistical indicator related to description length, which is hard to completely reflect the quality of the generated captions. Therefore, it is necessary to explore generative evaluation indicators such as CLIPScore[1] and GPTScore[2] that use pre-trained models.
2.	**The choices of VLMs are limited, which may lead to unobjective conclusions.** They just use the OpenFlamingo as the multi-modal learner.  The authors should add more VLMs to explore the in-context configurations for image captioning, e.g. MiniGPT4[3], LLaVA[2].

3.	**The observation is a little overclaiming.** The in-context configurations observation from the image caption is not equal to the VL in-context learning. There are many visually-conditioned VL tasks that need to be explored such as VQA[4], and dense image captioning.

4.    **The method section of iteratively prompting (IP) is confusing.** It is hard to understand how the algorithm of IP works, and the lack of analysis for IP.


[1] Hessel, Jack, et al. "Clipscore: A reference-free evaluation metric for image captioning." arXiv preprint arXiv:2104.08718 (2021).

[2] Liu, Haotian, et al. "Visual instruction tuning." arXiv preprint arXiv:2304.08485 (2023).

[3] Zhu, Deyao, et al. "Minigpt-4: Enhancing vision-language understanding with advanced large language models." arXiv preprint arXiv:2304.10592 (2023).

[4] Tsimpoukelli, Maria, et al. "Multimodal few-shot learning with frozen language models." Advances in Neural Information Processing Systems 34 (2021): 200-212.

**Questions:**

I tend to increase the review score once the following questions are answered well. The following questions are shown in weaknesses.

---

> ### Author Rebuttal · Authors · 2023-08-09
>
> **1. CLIPScore Assessment.**
> Upon your recommendation, we employ the CLIPScore to re-assess the quality of our generated captions. Due to space constraints, we present only the key findings here which validate two key conclusions shown in Lines 227-229 and 302-306. We first show the results about the first conclusion:
> | Image Selection | Caption Assignment | Mean CLIPScore |
> |-----------------|---------|------|
> | RS | GTC | 79.14 |
> | RS | MGC-TF@135 | 78.05 |
> | SIIR-CLIP | GTC | 80.46 |
> | SIIR-CLIP | MGC-TF@135 | 79.79 |
>
> Here, RS denotes Randomly Sampling, SIIR-CLIP denotes CLIP embedding Similarity-based Image-Image Retrieval, GTC denotes using the first ground-truth caption among 5 human-labelled ones, and MGC-TF@135 denotes using the model generated captions whose CIDEr score is 135.
> From the table, it's evident that under RS, GTC (79.14) outperforms MGC-TF@135 (78.05), while utilizing SIIR-CLIP causes MGC-TF@135 (80.46) surpassing GTC (79.79). This observation is consistent with the findings given in Lines 230-238, which also supports our claim in Lines 227-229.
>
> Then we show the results about the second one conclusion that similar images lead to short-cut inference, where we measure the captions whose CIDEr scores are reported in the following table:
> |   | In-Context Images | GTC CLIPScore | ICC  CLIPScore |
> |---|-------------------|------------------|------------------|
> |(1)  | Test Image        | 50.70            | 77.78            |
> |(2)  | SIIR-CLIP         | 53.50            | 77.44            |
> |(3)  | RS                | 73.90            | 59.03            |
>
> where we still observe that as in-context images more similar to the test image, VLM tends to mimic in-context captions (ICC), which is consistent with the findings in Lines 315-319. For instance, as the similarity from method (1) to (3) declines, the CLIPScore diverges more from ICC but converges towards ground-truth captions (GTC). More results using CLIPScore and GPTScore will be incorporated in revision.
>
> **2. LLaVa and MiniGPT.**
> LLaVa and MiniGPT are primarily designed for instruction tuning where it works in a multi-round chat manner instead of an in-context learning manner. The network architectures of both models are not feasible to handle extensive, interleaved vision and language data as Flamingo. Moreover, they lack the appropriate training losses to effectively handle few-shot prompts inputs in the manner that Flamingo does. As such, they may be capable of processing 4-shot inputs in the manner of multi-turn chat, while will fail to deal with much more shot data like 8 or 16-shots. However, to investigate the impact of diverse configurations, a model should be able to handle much more shot data. Considering these disadvantages, we do not use them to explore in-context configurations.
>
> However, we still try some other VLMs to explore in-context configurations. Please refer to **"7. Experiments on Smaller Open-flamingo and Otter."** in the response to **Reviewer tg5h** to see the results test on Otter and a smaller version of Open-Flamingo, which are two models with in-context learning ability published after the NeurIPS submission deadline.
>
> **3. VQA Results.**
> We follow your suggestion to explore another VL task to test the major conclusion of our study. Given our computational limitations and the tight rebuttal timeframe, we choose to explore VQAv2 since it is more different from image captioning compared to dense captioning, which is also suggested by Reviewer vuiF. To explore different configurations in VQA. We use two different image selection strategies which are Random Selection (RS) and  CLIP Similarity-based Image-Image Retrieval (SIIR-CLIP). Moreover, to ensure varying quality levels of in-context text—best, middle, and worst, we use matched (MAT), half-matched (HMAT), and mismatched (MMAT) question-answer pairs. Here, MAT, HMAT, and MMAT signify that all, half, or none of the answers are correct, respectively.
> | Image Selection | QA pairs     | Mean Accuracy |
> |-----------------|--------------|--------------|
> | RS              | MAT| 47.97|
> | RS              | HMAT| 47.34|
> | RS              | MMAT| 47.13|
> | SIIR-CLIP | MAT| 48.48|
> | SIIR-CLIP | HMAT| 47.55|
> | SIIR-CLIP | MMAT| 46.95|
>
>
> We can find that when the text quality is high, i.e. using matched question-answer pairs (MAT), we have SIIR-CLIP > RS, suggesting that using similar images achieves higher accuracy than random sampling. However, when using low text quality, e.g., HMAT or MMAT, similar images do not always lead to better results, e.g., with MMAT, SIIR-CLIP < RS. Such observation is consistent with the finding in Lines 230-238 and supports the major conclusion that image and text selection strategies influence each other.
>
> **4. Iterative Prompting (IP).**
> Briefly, IP employs VLM for iterative in-context caption generation. It is designed for scenarios with abundant images but limited human-labelled captions. Consider $\mathcal{D}=\{(I_1,C_1);...;(I_n,C_n), \tilde{I}_1;...;\tilde{I}_N\}$ where $n<<N$ (e.g., $n=32$ and $N=10000$) and only $n$ images contain human-labelled captions. Given a test image $\hat{I}$, traditional image selection becomes impractical due to the scarcity of available captions. Therefore, we suggest generating a caption $\tilde{C}$ for each $\tilde{I} \in \mathcal{D}$ using Eq.(1) where $\mathcal{S}=\{(I_1,C_1);...;(I_n,C_n)\}$. This provides each image in $\mathcal{D}$ with a corresponding caption, allowing the use of earlier image selection methods like SIIR or SICR (Similarity-based Image-Caption Retrieval). This process can be iteratively executed, hence the name Iterative Prompting.
>
> In essence, MGC-VLM(N) introduced in Line 169 is equal to employing this method once. Given the in-depth analysis of the captions of MGC-VLM(N) in section 4.2.1, we bypass their re-examination in lines 274-286 but try to know how many iterations IP needs to make the performance saturate. We will clarify this in the revision.

---

### Decision · Program_Chairs · 2023-09-21

**Decision:**

Accept (poster)

**Comment:**

This work received unanimously positive rates. The final version may need to involve the rebuttal.